

# Numerical signatures of ultra-local criticality in a one dimensional Kondo lattice model

Alexander Nikolaenko[1] and Ya-Hui Zhang[2*]

**1** Department of Physics, Harvard University, Cambridge, MA, USA
**2** Department of Physics and Astronomy, Johns Hopkins University, Baltimore, MD, USA

* yzhan566@jhu.edu

## Abstract

Heavy fermion criticality has been a long-standing problem in condensed matter physics. Here we study a one-dimensional Kondo lattice model through numerical simulation and observe signatures of local criticality. We vary the Kondo coupling $J_K$ at fixed doping $x$. At large positive $J_K$, we confirm the expected conventional Luttinger liquid phase with $2k_F = \frac{1+x}{2}$ (in units of $2\pi$), an analogue of the heavy Fermi liquid (HFL) in the higher dimension. In the $J_K \leq 0$ side, our simulation finds the existence of a fractional Luttinger liquid (LL*) phase with $2k_F = \frac{x}{2}$, accompanied by a gapless spin mode originating from localized spin moments, which serves as an analogue of the fractional Fermi liquid (FL*) phase in higher dimensions. The LL* phase becomes unstable and transitions to a spin-gapped Luther-Emery (LE) liquid phase at small positive $J_K$. Then we mainly focus on the 'critical regime' between the LE phase and the LL phase. Approaching the critical point from the spin-gapped LE phase, we often find that the spin gap vanishes continuously, while the spin-spin correlation length in real space stays finite and small. For a certain range of doping, in a point (or narrow region) of $J_K$, the dynamical spin structure factor obtained through the time-evolving block decimation (TEBD) simulation shows dispersion-less spin fluctuations in a finite range of momentum space above a small energy scale (around $0.035J$) that is limited by the TEBD accuracy. All of these results are unexpected for a regular gapless phase (or critical point) described by conformal field theory (CFT). Instead, they are more consistent with exotic ultra-local criticality with an infinite dynamical exponent $z = +\infty$. The numerical discovery here may have important implications on our general theoretical understanding of the strange metals in heavy fermion systems. Lastly, we propose to simulate the model in a bilayer optical lattice with a potential difference.

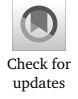

# 1  Introduction

The study of quantum phase transition between a small Fermi surface phase and a large Fermi surface phase is a central topic in modern quantum condensed matter physics and may be closely related to the strange metals observed in heavy Fermion systems [1–8] and in hole-doped high Tc cuprates [9–12]. The standard Landau-Ginzburg theory involves the onset of a symmetry-breaking order and its fluctuation [13,14]. However, a number of experiments in heavy Fermion systems [15–17] do not appear to be consistent with the simple spin-density-wave (SDW) approach. It was suggested that the transition in heavy fermion systems may be characterized by a jump in Fermi surface volume resulting from Kondo breakdown, rather than fluctuations in symmetry-breaking orders. There have been many attempts to formulate a framework of an exotic transition following different approaches, such as extended dynamical mean field theory (EDMFT) [18], fractionalization and slave boson theory [19–21], ancilla qubit theory [22,23]. However, a well-established theoretical description of such a Kondo breakdown transition is still elusive.

In this paper, we take a microscopic approach to avoid uncontrolled approximations usually existing in low-energy effective field theory methods. Specifically, we will numerically simulate a one-dimensional Kondo lattice model using density matrix renormalization group (DMRG) [24]. DMRG has been demonstrated to be an unbiased method with excellent performance in one dimension (1D). Therefore, the numerical results should be reliable. The only question is whether there is anything interesting in a 1D model. We will show that the answer is yes and we find a critical point or phase which seems to support local criticality behaviour. We note

that there already exist a few numerical studies of the Kondo lattice model in one dimension [25–28], but to our best knowledge, there is no detailed study of how a Kondo breakdown phase at negative $J_K$ evolves to the Luttinger liquid in the large positive $J_K$ at a generic filling.

The model we study consists of a $t$-$J$ model of itinerant electron and a Heisenberg model of spin 1/2 chain [29]. They couple to each other through a Kondo coupling $J_K$. At a density $x$ for the itinerant electron, we vary $J_K$ to study the phase diagram. In the $J_K \leq 0$ side, the ground state has one charge mode and two spin modes (C1S2), where the localized spin 1/2 moments provide an additional gapless mode with momentum $Q = \pi$. The itinerant electron forms a Luttinger liquid with $2k_F^* = \frac{x}{2}$ (in units of $2\pi$). The phase is an analogue of the fractional Fermi liquid (FL*) phase in higher dimension and we call it fractional Luttinger liquid (LL*) [30]. In the large positive $J_K$ we find the expected Luttinger liquid (LL) phase with $2k_F = \frac{1+x}{2}$ (in units of $2\pi$), which is an analogue of the heavy Fermi liquid (HFL) phase in the higher dimensional Kondo lattice model. Therefore, we have the same problem of small to large Fermi surface evolution as in higher dimensions. Complexity arises in one dimension because the LL* phase is unstable at small positive $J_K$ and transitions to a Luther-Emery liquid (LE) phase with a spin gap and only one gapless charge mode [26, 29, 31, 32]. The LE phase is best described as a descendant of the LL* phase [29]. It is similar to a superconductor phase in a higher dimension and above the energy scale of the spin gap it smoothly connects to the LL* phase. We note, that in the heavy Fermion experiments, the transitions between the small and large Fermi surface metals are typically covered by a superconductor dome. Thus, the situation in 1D is similar to higher dimension and we will try to understand the nature of the evolution from the LE phase to the LL phase upon increasing $J_K$. The hope is that there may also be a 'strange metal' critical point or a phase in between.

As the LE phase descends from the LL* phase and we are not aware of any way to construct it from the LL phase, we do not expect any obvious continuous transition between the LE and LL phases. Indeed, we find that there is either a first-order transition or an intermediate region in between. We will focus on the latter case and provide evidence of local criticality behaviour beyond the familiar Luttinger liquid or conformal field theory (CFT) descriptions. At one point (or a narrow region) of $J_K$, we find that the spin gap is almost vanishing, while there is still a finite correlation length in equal time spin-spin correlation function in real space. Meanwhile, the dynamical spin structure factor $S(\omega, q) \sim \text{Im}\chi_S(\omega, q)$ obtained from the time-evolving block decimation (TEBD) simulation shows dispersion-less spin fluctuations in a range of the momentum space above an energy cutoff (around $0.035J$, $J$ is the Heisenberg spin coupling) imposed by the numerical accuracy itself. Such behaviour resembles what is called local criticality. We note, that in the literature sometimes local criticality is also used [18] for the case where only the self-energy is momentum independent, while there is still a significant spatial correlation. In this weaker case, the dynamical exponent is still finite. The behaviour in our model is closer to a stronger definition with an infinite dynamical exponent. Therefore, we follow Ref. [33] and call it ultra-local criticality to be distinguished from the weaker definition.

The discovery of ultra-local critical spin fluctuations above a small energy scale is quite remarkable, as this phenomenon is not generally believed to be possible in a reasonable model with translation invariance and a finite-dimensional Hilbert space at each site. The existence of ultra-local criticality also has significant implications for our understanding of the strange metal. For example, it may be a loophole of the anomaly approach of non-Fermi liquid [33] and it is known that ultra-local critical spin fluctuations with a constant spectral function over frequency can lead to a marginal Fermi liquid and linear T resistivity [34]. On the experimental side, similar local critical behaviours have been discovered in neutron scattering measurements of some heavy Fermion materials [17,35]. One may worry that the experimental results arise from disorder effects. Our numerical observation of similar local critical behaviours in a clean model strongly suggests that such a phenomenon may likely be intrinsic and does not need

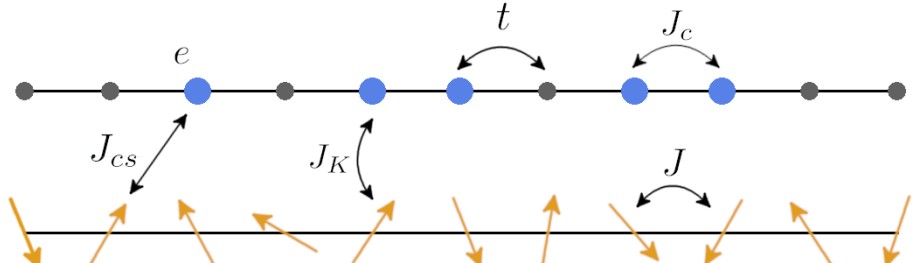

Figure 1: The geometry and corresponding couplings of the Hamiltonian in Eq. 1. The first layer corresponds to a *t-J* model, while the second layer is an antiferromagnetic spin 1/2 model. The two layers are coupled together through the on-site Kondo coupling $J_K$ and nearest neighbour Kondo interaction $J_{cs}$.

disorders. On the theoretical side, similar behaviour has been discussed in holographic theory from the gravity side and dubbed as 'semi-local quantum fluid' [36]. However, we are not aware of a well-established theory of ultra-local criticality for a local and translation invariant quantum lattice model directly. We hope our numerical confirmation of the existence of ultra-local criticality will stimulate theoretical efforts in this direction. Lastly, we propose to simulate the Kondo lattice model in a bilayer optical lattice with a potential difference, which hopefully will provide more information at finite temperatures and higher dimensions.

## 2 Model and phase diagram

We formulate our model as a generalization of the Kondo-Heisenberg lattice model, which is described by the following Hamiltonian:

$$
H = -tP \sum_{<i,j>,\sigma} (c_{i,\sigma}^{\dagger} c_{j,\sigma} + h.c)P + J_c \sum_{\langle ij \rangle} \vec{S}_i^e \cdot \vec{S}_j^e + (V - \frac{1}{4} J_c) \sum_{\langle ij \rangle} n_i n_j
$$
$$
+ J \sum_{<i,j>} \vec{S}_i \cdot \vec{S}_j + J_K \sum_i \vec{S}_i^e \cdot \vec{S}_i + J_{cs} \sum_{\langle ij \rangle} \vec{S}_i^e \cdot \vec{S}_j + \vec{S}_i \cdot \vec{S}_j^e . \tag{1}
$$

The first layer is described by the *t-J* model, $P$ is the projection operator to forbid double occupancy and $\vec{S}_i^e = \frac{1}{2} \sum_{\sigma\sigma'=\uparrow,\downarrow} c_{i;\sigma}^{\dagger} \vec{\sigma}_{\sigma\sigma'} c_{i;\sigma'}$ is the spin operator of the itinerant electron. The couplings $V$ and $J_c$ account for the nearest neighbour interaction in the first layer. The second layer is described by Heisenberg spin 1/2 model with coupling $J$. We will call these two layers C layer and S layer respectively in what follows. Finally, we have the inter-layer Kondo couplings $J_K$ and $J_{cs}$. Fig. 1. shows the geometry and the corresponding couplings pictorially. In the Appendix A we show how the studied model can be realized in bilayer optical lattices.

In the rest of the paper, we fix $t = 1$, $J = J_c = 0.5$ and study how the system evolves as we change Kondo coupling $J_K$. We also use two different values $J_{cs} = 0, 0.5J$ which we concluded does not change the underlying physics much. We will simulate the model with both finite

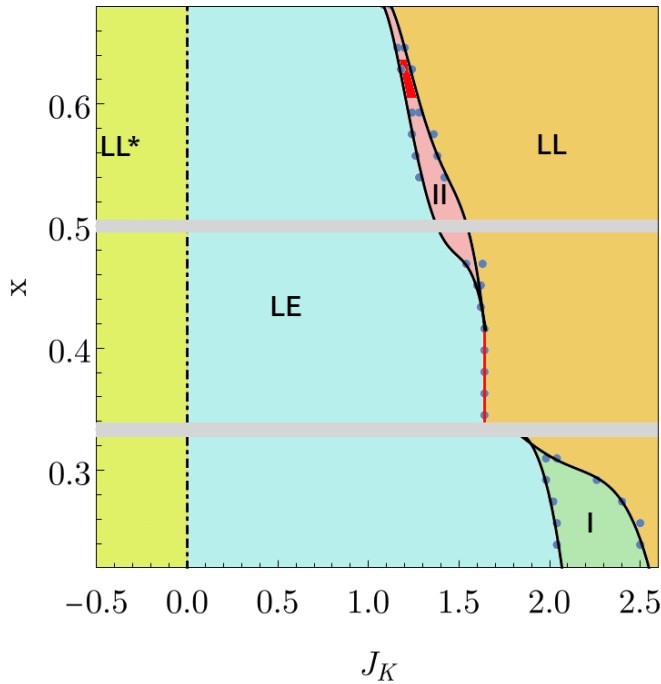

Figure 2: Illustration of phase diagram of the Kondo lattice model with $J_{cs} = 0.5J, V = 4J$. LL* phase corresponds to fractional Luttinger liquid, LE stands for Luther-Emery(spin gap) phase, and LL is a Luttinger liquid phase. LL*, LE and LL phases can be labeled as C1S2, C1S0 and C1S1 respectively and they have central charges $c = 3, 1, 2$. Here CmSn means that there are m charge modes and n spin modes. Grey shadowed regions correspond to commensurate fillings $x = 1/3$ and $x = 1/2$ where the system turns into a charge density wave (CDW) insulator. The red vertical line marks the first-order transition between LE and LL phases. Region I is a gapless phase with a central charge $c = 3$. When approaching the region I from the LE phase, the spin gap vanishes continuously, while the spin correlation length in real space stays finite and small, indicating possible infinite dynamical exponent. Region II hosts an exotic phase with a weak ferromagnetic moment and ultra-local criticality at the phase boundary. Within region II, around the doping $x \approx 0.61-0.63$, there is a re-entrance of another spin-gapped phase. We find signatures of ultra-local criticality between the two spin-gapped domes. We use system size $L = 113$, and maximum bond dimension $m = 1000$ with finite DMRG for this plot.

and infinite DMRG. The bond dimension varies from 500 to 8000 depending on parameters. The typical truncation error is at order $10^{-8}$ or even smaller.

We start our analysis by providing an illustrated phase diagram of the model in Fig. 2. Previous calculations have found a dominant ferromagnetic phase in the conventional Kondo lattice model with $J = 0$ [25]. Here we use $J = 0.5t$ to get rid of the FM order. Then the phase diagram is dramatically different from that of the conventional Kondo lattice model with $J = 0$.

At $J_K = 0$, we can start from the layer decoupled phase. We know the itinerant electron in the C layer just forms a spinful Luttinger liquid, while the spin moments in the S layer form a gapless phase with one spin mode. We can dub this phase C1S2 because it has one charge mode and two spin modes. The itinerant electrons in the C layer form a Fermi surface with $2k_F^* = \frac{x}{2} \times 2\pi$, which is different from the required value of the Luttinger theorem by $1/2$ of $2\pi$. This feature is similar to the fractional Fermi liquid (FL*) phase discussed in higher dimensions. Therefore, we dub this phase a fractional Luttinger liquid (LL*) [29]. The LL* phase is stable

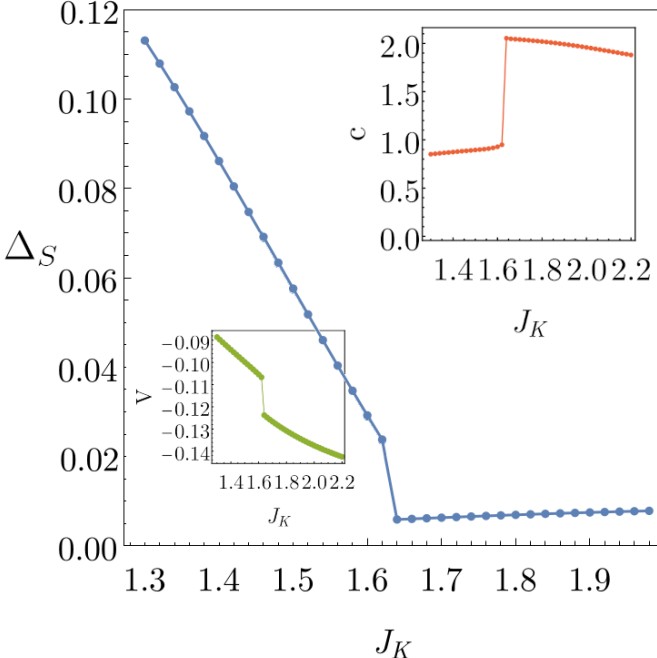

Figure 3: Spin gap $\Delta_S$ jumps at the first order transition. $L = 113$, $x = 45/113$ and maximum bond dimension $m = 1000$. In the inset, we also show the jump of $V = 2\langle \vec{S}_i \cdot \vec{S}_i^e \rangle$ and the central charge $c$. We use $J_{cs} = 0.5$ and $V = \frac{1}{4}J_c$ for this plot.

in the negative $J_K$ regime. However, it is unstable to a spin-gapped Luther Emery (LE) liquid phase with a finite positive $J_K$ [29]. In the large positive $J_K$, we recover the Luttinger liquid (LL) as an analogue of the heavy Fermi liquid in higher dimensions. The LL phase has a Fermi surface with $2k_F = \frac{1+x}{2} \times 2\pi$, satisfying the Yamanaka-Oshikawa theorem [37]. Note that the central charge for the LL*, LE, and LL phases are $c = 3, 1, 2$ respectively and they can be labeled as C1S2, C1S0, C1S1.

Although the LL* phase is unstable to the spin-gapped LE phase, one can view the LE phase as a descendant of the LL* phase. Above the energy scale of the spin gap, we can still think of this phase as a LL* phase with a small Fermi surface. Therefore we can ask how the small Fermi surface changes to the large Fermi surface in the large $J_K$ regime. In the regime of intermediate filling $x \in (0.33, 0.43)$ the transition appeared to be of the first order, labeled as the red line in Fig. 2. As evidence of the first order transition, the spin gap $\Delta_s$ jumps to zero discontinuously and other physical quantities such as $V = 2\langle \vec{S}_i \cdot \vec{S}_i^e \rangle$ also experience a jump, as shown in Figure 3. The central charge changes from $c = 1$ in the LE phase to $c = 2$ in the LL phase directly at the transition.

## 2.1 Intermediate region I

At small doping $x < \frac{1}{3}$ the LE phase evolves to the LL phase through an intermediate region I. Region I has a central charge $c = 3$ and a finite spin susceptibility, in agreement with a conformal field theory (CFT) description with both gapless charge and spin modes. We list results for intermediate region I at $x = \frac{7}{31}$, $J_{cs} = 0.5J$ in Fig. 4. In Fig. 4(a) we plot $\Delta_S L$ from finite DMRG, where $\Delta_S$ is the spin gap and $L$ is the system size. $\Delta_S$ is obtained from $E(S_z^t = 1) - E(S_z^t = 0)$, where $E(S_z^t = m)$ is the ground state energy of the sector of the total spin $S_z = m$ sector. Note that the total $S_z^t = S_z + S_z^e$ component is conserved in our calculation, so we can target a state at each $S_z^t = m$. It is known that the inverse of the uniform spin

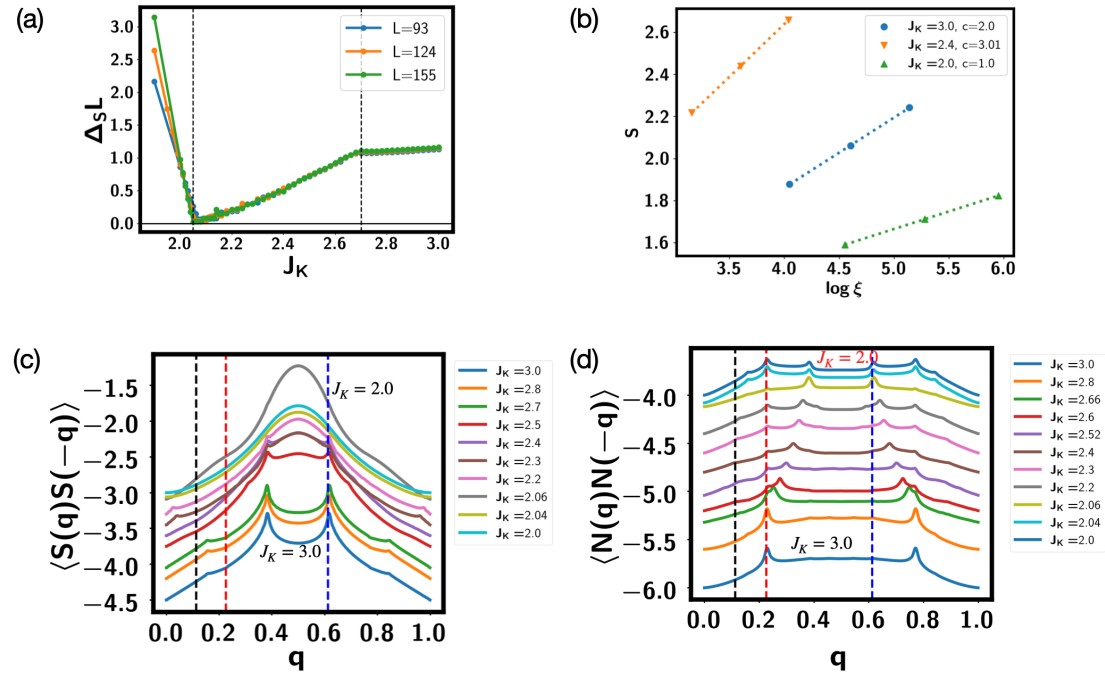

Figure 4: Results for the intermediate region I at $x = \frac{7}{31}$, $J_{cs} = 0.5J$, $V = 0$. (a) $\Delta_S L$ for a few system sizes obtained from finite DMRG with bond dimension $m = 2000$. $L$ is the system size and $\Delta_S$ is the spin gap. $\Delta_S L$ is proportional to the inverse of the uniform spin susceptibility. The two dashed lines are at $J_K = 2.05$ and $J_K = 2.7$, which mark the phase boundaries of the intermediate region I. (b) Central charge fit from infinite DRMG with unit cell $L = 31$. The central charge is $c = 1, 3, 2$ for the three phases when increasing $J_K$. (c) Spin-spin correlation function in momentum space. Here we use the total spin operator $\vec{S^t} = \vec{S} + \vec{S^e}$. (d) Density-density correlation function in momentum space. In (c)(d) the black dashed vertical line labels $q = 2k_F^* = \frac{x}{2} \times 2\pi$. The red dashed line labels $q = 4k_F = x \times 2\pi$. The blue dashed line labels $q = 2k_F = \frac{1+x}{2} \times 2\pi$. In (c)(d) the lines of different $J_K$ are shifted, so the absolute value of the y-axis is meaningless.

susceptibility $\chi_S^{-1} \propto \Delta_S L$. When $J_K < 2.05$, we can see that $\Delta_S L$ increases with the system size $L$, indicating a finite spin gap in agreement with the LE phase. But when $J_K > 2.05$, $\Delta_S L$ is constant with system size, indicating a finite uniform spin susceptibility. This is expected from the scaling $\Delta_S \sim \frac{1}{L}$ of a conformal field theory (CFT) description.

Using bond dimension $m$ from 500 to 2000 we fit the central charge with the formula $S = \frac{c}{6} \log \xi$, where $\xi$ is the correlation length obtained from the transfer matrix technique [38] and $S$ is the entanglement entropy. Inside the intermediate region I, we find that the central charge is $c = 3$ from the infinite DMRG result in Fig. 4(b). This central charge is larger than both the LE phase ($c = 1$) on the left and the LL phase ($c = 2$) on the right. One natural interpretation is that there are two Fermi surfaces per spin component in the intermediate region I, leading to two charge and two spin modes. Then one of the four modes gets gapped, giving $c = 3$. In the Appendix G, we will argue that a simple mean-field theory is able to explain the existence of several Fermi surfaces and show how a flat band scenario is able to explain a finite correlation length in the gapless system.

To support the above picture, we indeed find that the peak of the spin-spin correlation function $\langle \vec{S}(q) \cdot \vec{S}(q) \rangle$ is still at $2k_F = \frac{1+x}{2}$ (in units of $2\pi$) in the intermediate region (see Fig. 4(c)), while the peak of density-density correlation functions $\langle N(q)N(-q) \rangle$ shifts from $2k_F$ to $4k_F$ gradually in the intermediate phase I, as shown in Fig. 4(d). A gradually changing momentum is a signature of a split Fermi surface. Based on the value of the central charge, the phase could be either C1S2 or C2S1. We conjecture that it is C2S1 and there is only one spin mode, given that the peak of the spin-spin correlation function seems to be pinned at $2k_F$. But more analysis is needed to fully understand how a spin mode gets gapped starting from four modes. Except for the unusually odd central charge, the phase is otherwise consistent with a CFT. It easily converges in our numerical calculation with expected CFT behaviour.

## 2.2 Dip of inverse charge compressibility and Luttinger parameter

Overall at a generic filling, we find dips in both the inverse charge compressibility $\kappa_c^{-1}$ and the Luttinger parameter $K_c$ in the intermediate regime, shown in Fig. 5. They are extracted using the formulas $\kappa_c^{-1} = \partial\mu/\partial n = L(E(N+2) + E(N-2) - 2E(N))/4$ and $\langle N(q)N(-q) \rangle = K_c q/2\pi$ at small $q$. We find $K_c < \frac{1}{3}$ quite generically, indicating strong repulsive interaction. Given that $\kappa_c = \frac{\pi K_c}{v_c}$ with $v_c$ as the charge velocity in a Luttinger liquid, a dip of both $\kappa_c^{-1}$ and $K_c$ means that the velocity $v_c$ goes down even faster than $K_c$. This also means that the Drude weight $D_c \propto K_c v_c$ gets much smaller in the intermediate region. All of these properties suggest that the intermediate region has a large repulsive interaction and slow charge velocity. It is a region where the charge compressibility tends to become large, while the Drude weight tends to vanish.

## 2.3 Charge density wave at commensurate filling

One consequence of the small Luttinger parameter $K_c$ is that the ground states at commensurate filling such as $x = \frac{1}{3}, \frac{1}{2}$ are charge density wave(CDW) insulators. That is because the umclapp terms become relevant for a small Luttinger parameter. To identify the insulating nature, we computed the inverse charge compressibility $\kappa_c^{-1}$. In the insulating phase we have $\kappa_c^{-1} = L\Delta_c/2$ which means that inverse compressibility diverges when $L \to \infty$. As shown in the inset of Fig. 5(b), at commensurate filling $\kappa_c^{-1}$ is significantly larger than the corresponding one at incommensurate filling nearby. Moreover, in the insulator phase we expect that $\langle N(q)N(-q) \rangle \sim q^2$ at small $q$. This is indeed the case as shown in the inset of Fig. 5(a).

# 3 Unconventional criticality around the region I

After we have a general understanding of the global phase diagram in the $(J_K, x)$ parameter space, we now zoom in on the 'critical region' to understand the evolution from the LE phase to the LL phase. As shown in the phase diagram, we never find a direct continuous transition between the LE phase and the LL phase. Instead, we find either a first-order transition or another intermediate phase. The intermediate phase I appears to be well described by a CFT with $c = 3$. Below we are interested in how the spin gap closes when approaching this intermediate phase I, starting from the Luther-Emery liquid phase at small $J_K$.

Surprisingly we find that the transition between the Luther-Emery liquid phase and the $c = 3$ intermediate phase (in Region I of Fig. 2) is not described by a usual conformal field theory(CFT). First, when approaching the critical point between the LE and the intermediate region I around $J_K = 2.05$, $\Delta_s L$ vanishes and the uniform spin susceptibility diverges, shown in Fig. 4(a). This is already unexpected from a usual critical point described by CFT, where we should still expect $\Delta_S \propto \frac{1}{L}$ and a finite $\Delta_S L$.

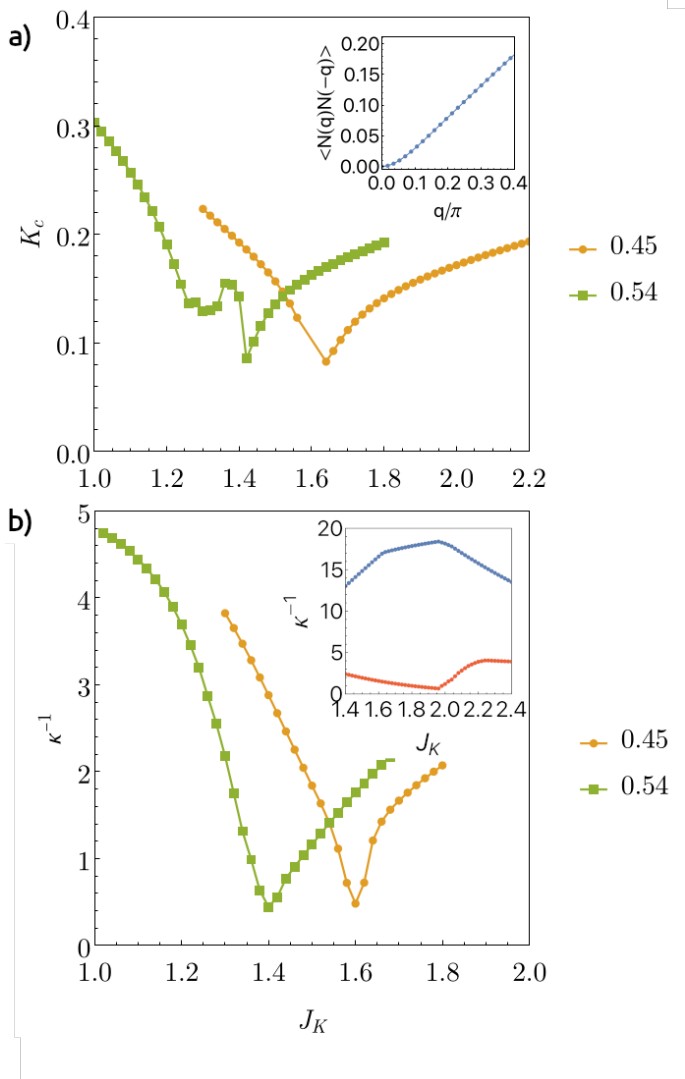

Figure 5: a) Charge Luttinger parameter for different dopings $x$. The inset shows density-density correlations at small $q$ at commensurate doping $x = 37/113 \approx 0.33$ to demonstrate the quadratic behaviour. b) Inverse compressibility for different dopings $x$. The inset shows inverse compressibility for a commensurate doping $x = 37/113 \approx 0.33$(blue line) and incommensurate $x = 35/113 \approx 0.31$(red line). The maximum bond dimension $m = 1000$(finite DMRG).

Besides, when approaching the critical point from the spin-gapped phase, the correlation length remains finite, shown in Fig. 6. At $J_K = 2.05$, we still have a very small correlation length ( $\xi_S \approx 2$) in spin channel corresponding to $\xi_S^{-1} \approx 0.46$. Then across the critical point, $\xi_S^{-1}$ jumps to 0. This is a clear signature that the dynamical exponent $z$ at the critical point must be larger than 1, because otherwise in a relativistic critical theory we should expect the inverse spin correlation length $\xi_S^{-1} \propto \Delta_S$ and also vanishes from the spin gapped side.

In summary, when approaching the critical point $J_K^c \approx 2.05$ from the LE phase, we find that the spin gap $\Delta_S$ goes to zero continuously, indicating a divergent correlation length $\xi_t$ in the time direction. But the correlation length $\xi_S$ in the real space stays finite (around 2 even at $J_K \to J_K^c - \epsilon$, with $\epsilon$ an infinitesimal number). Then if we use the conventional scaling $\xi_t \sim \xi_S^z$, we reach a striking conclusion that $z = +\infty$. At small $J_K$ the spin excitation has a dispersion $\omega^2 = c_s^2(\delta q)^2 + \Delta^2$ with $\delta q = q - \pi$. Initially $\Delta$ increases with $J_K$, but then $\Delta$ decreases when

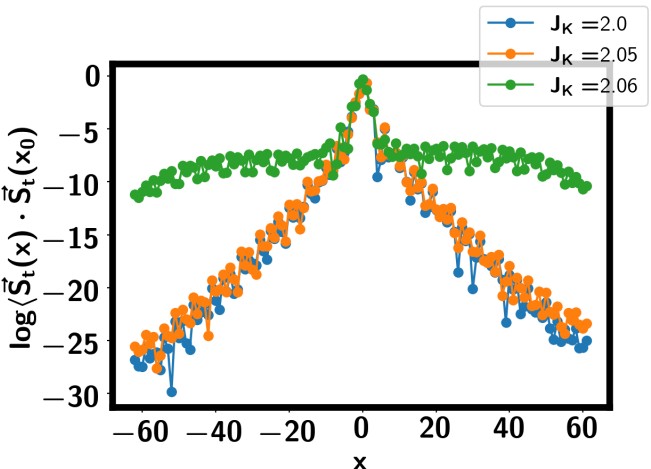

Figure 6: Spin-Spin correlation functions when approaching the critical point $J_K = 2.05$ from the spin-gapped Luther Emery liquid phase. At $J_K = 2.05$, we still have a finite correlation length $\xi_S^{-1} \approx 0.46$. Then $\xi_S^{-1}$ jumps to 0 into the intermediate phase with $c = 3$. The jump of the correlation length from finite to infinite around $J_K^c \approx 2.05$ is in contrast to the continuous vanishing of $\Delta_S L$ in Fig. 4(a), indicating a critical point with dynamical exponent $z > 1$, and probably $z = +\infty$. The parameters are the same as in Fig. 4 with system size $L = 124$ in finite DMRG.

$J_K$ is close to the critical point $J_K^c \approx 2.05$. Note in this ansatz the inverse of the real space spin correlation length is $\xi_S^{-1} = \Delta/c_s$. So the only way that $\xi_S^{-1}$ can stay finite is that the velocity $c_s$ also vanishes along with the gap $\Delta$. So we are in an unusual situation: the gap is closing while the dispersion of the excitation also becomes flat. We leave it to the future to develop an analytical theory of this kind of exotic criticality.

# 4 Evidence of ultra-local criticality around region II

The intermediate region I in Fig. 2 seems to be well described by a CFT. In contrast, the intermediate region II (see Fig. 2.) is much more exotic.

In the following, we provide numerical evidence for ultra-local criticality around region II with dynamical exponent $z = +\infty$. We will also show evidence of gapless spin fluctuations in a range of momentum space in the dynamical spin structure factor.

Inside region II, there is a small sub-region coloured red in Fig. 2. This small regime has a re-entrance of a spin gap. In the other places of region II, there is no spin gap. Instead, the inverse spin susceptibility even vanishes. We will show that it has a weak ferromagnetic moment and/or spin glass behaviour. We will discuss these two cases separately. Both of them have signatures of ultra-local criticality at a critical point (or region) of $J_K$.

## 4.1 Ultra-local criticality between LE and LE₂ phase

We first look at the subregion inside region II coloured red in Fig. 2. We list results in Fig. 7 at $x = \frac{7}{11}$ for $J_{cs} = 0$. Fig. 7(a) shows two spin-gapped phases with central charge $c = 1$, which we call LE and LE₂. In between them, the central charge seems to approach 2 (see Fig. 7(c)). We use the relationship $S = \frac{c}{6} \log \xi$, where $\xi$ is the correlation length in the charge sector $(Q, S_z) = (0, 0)$, which serves as an effective length scale of the infinite DMRG. We then

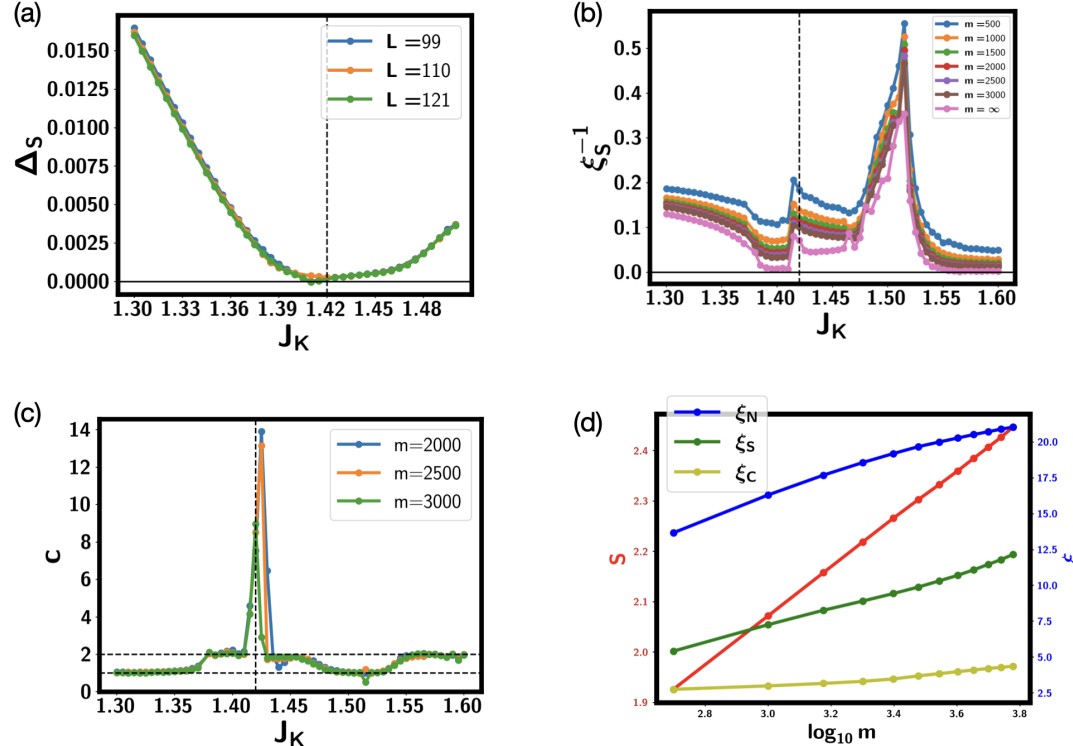

Figure 7: Numerical results at $J_{cs}=0$ with $x=\frac{7}{11}$. (a) Spin gap $\Delta_S$ from finite DMRG with system size $L = 99, 110, 121$. The bond dimension is $m = 2000$. The dashed line is at $J_K = 1.42$. One can see that there is a re-entrance of spin gapped phase when $J_K > 1.42$. (b) The inverse spin correlation length $\xi_S^{-1}$ obtained from infinite DMRG for bond dimension $m$ up to 3000. We use a unit cell size $L = 22$. (c) Central charge from infinite DMRG. The two dashed horizontal lines label $c = 1, 2$. (d) The growth of the entanglement entropy $S$ and the correlation length $\xi$ with the bond dimension $m$. We also plot the correlation length $\xi_S$ in the sector $(Q, S_z) = (0, 1)$ and $\xi_C$ in the sector $(Q, S_z) = (1, \frac{1}{2})$. One can see that the single electron correlation length $\xi_C$ is around 3 as in finite DMRG. For the spin correlation length $\xi_S$, it reaches $\xi_S \approx 12.5$ for $m = 6000$, only slightly larger than the value from finite DMRG (see Fig. 8 below).

get $c(m) = 6\frac{\partial S}{\partial \log \xi}$ where the derivative is calculated with the values from two nearby bond dimensions. There is one point of $J_K = 1.42$ of particular interest. The entanglement entropy at this point is growing faster than $\log \xi$ in infinite DMRG so one can not extract a reasonable central charge. As shown in Fig. 7(d), the entanglement entropy $S$ scales as $S \sim \log m$ with the bond dimension $m$, as in a usual CFT. However, the correlation length $\xi_N$ (obtained in the sector $(Q, S_z^t) = (0, 0)$) has a tendency of saturation with $\log m$. This indicates deviation from CFT behaviour at $J_K = 1.42$.

Furthermore, we analyzed the behaviour of spin correlation length $\xi_S$, see Fig. 7(b). We discovered that along with two spin-gapped phases, $\xi_S$ is finite at $J_K = 1.42$. $\xi_S$ is obtained from the transfer matrix technique in the charge sector $(Q, S_z^t) = (0, 1)$. The value at $m = \infty$ is extrapolated with the formula $\xi_S^{-1}(m) = \xi_S^{-1}(m = \infty) + a\frac{1}{m} + b\frac{1}{m^2}$. In Fig. 8 we additionally obtained the correlation length from finite DMRG at $J_K = 1.42$. We confirm that the spin correlation length $\xi_S \approx 10$. We also discover that $\xi_S$ becomes shorter with a larger bond dimension (see the Appendix C). The single electron Green function has an even shorter cor-

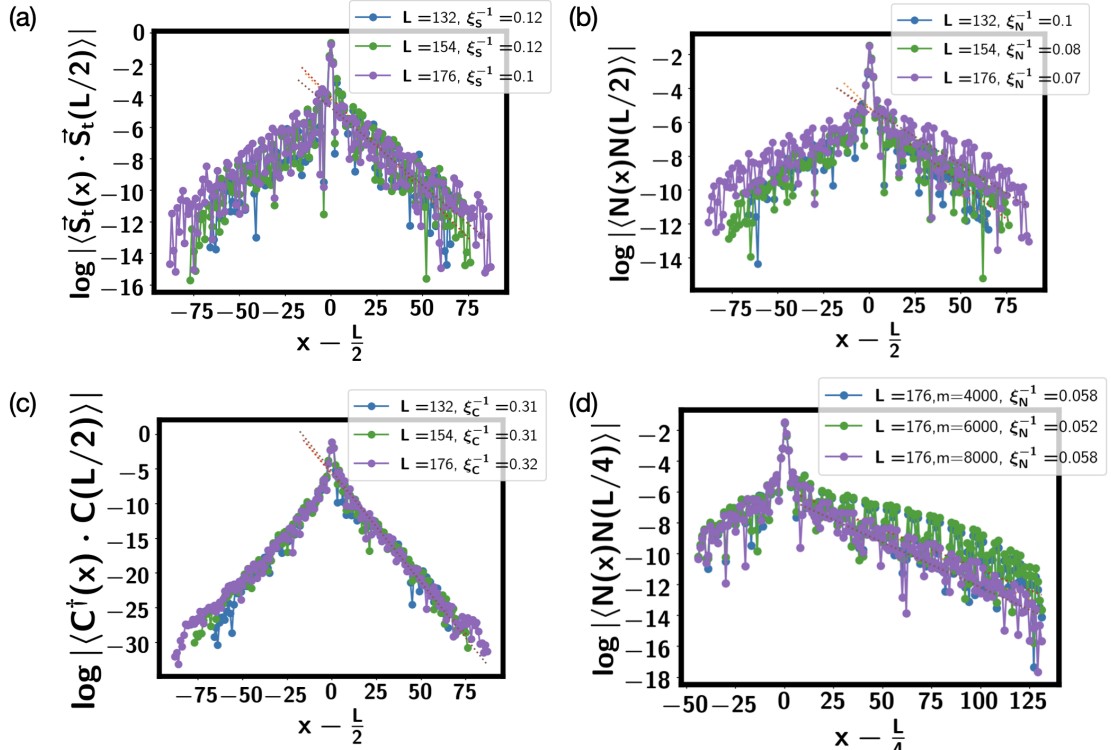

Figure 8: (a)(b)(c) Correlation function and fitted correlation lengths in finite DMRG with system size $L = 132, 154, 176$ for $J_K = 1.42$ at filling $x = \frac{7}{11}$. The bond dimension is $m = 6000$. (d) Evolution of the density-density correlation function with the bond dimension at $L = 176$. Here we fix $x_0 = L/4$. One can see that the correlation length $\xi_N$ becomes shorter with increasing bond dimension, suggesting a finite correlation length also in the density channel.

relation length of around 3 (see Fig. 8(c)). The correlation length in the density channel $\xi_N$ also appears to be finite with $\xi_N \approx 20$ from Fig. 8(d). This is also consistent with the infinite DMRG result in Fig. 7(d). In summary from both finite and infinite DMRG, we find that the correlation lengths in single electron, spin and density channels are all finite, which are around 3, 10 and 20 respectively.

Here we will mainly focus on the spin channel. A finite correlation length $\xi_S \approx 10$ is in contradiction with a vanishing spin gap (see Fig. 7(a)) at $J_K = 1.42$ if we assume an usual relativistic scaling $\Delta_S \propto \xi_S^{-1}$ with dynamical exponent $z = 1$. This suggests a dynamical exponent $z > 1$ in the spin-spin correlation. To further check the dynamical exponent, we plot the imaginary part of the dynamical spin susceptibility $\text{Im}\chi_{+-}(\omega, q)$ in Fig. 9, which is proportional to dynamical spin structure factor. The results are obtained from the TEBD algorithm (see the Appendix D for details). We apply the operator $S^-(L/2)$ to the ground state and then evolve the system under $e^{-iHt}$ to obtain $\langle S^\dagger(x, t)S^-(L/2, 0)\rangle$. $\text{Im}\chi_s(\omega, q)$ is then calculated by Fourier transformation. The total evolve time is $T = 100$ with a step $\delta t = 0.15$ for each TEBD step. The maximal bond dimension is set to be $m = 500$ in the calculation. In Fig. 9(a)(b) we get the expected spectroscopy results for the LL and LL* phase. One can see the gapless mode at $2k_F = \frac{1+x}{2} \times 2\pi$ for the LL phase and the gapless mode at $2k_F^* = \frac{x}{2} \times 2\pi$ for the LL* phase at $J_K = 0$. For the LL* the dominant spectral weight is at the momentum $Q = \pi$ from the local spin moments. Around the gapless momentum, the dispersion is linear in agreement with a dynamical exponent $z = 1$.

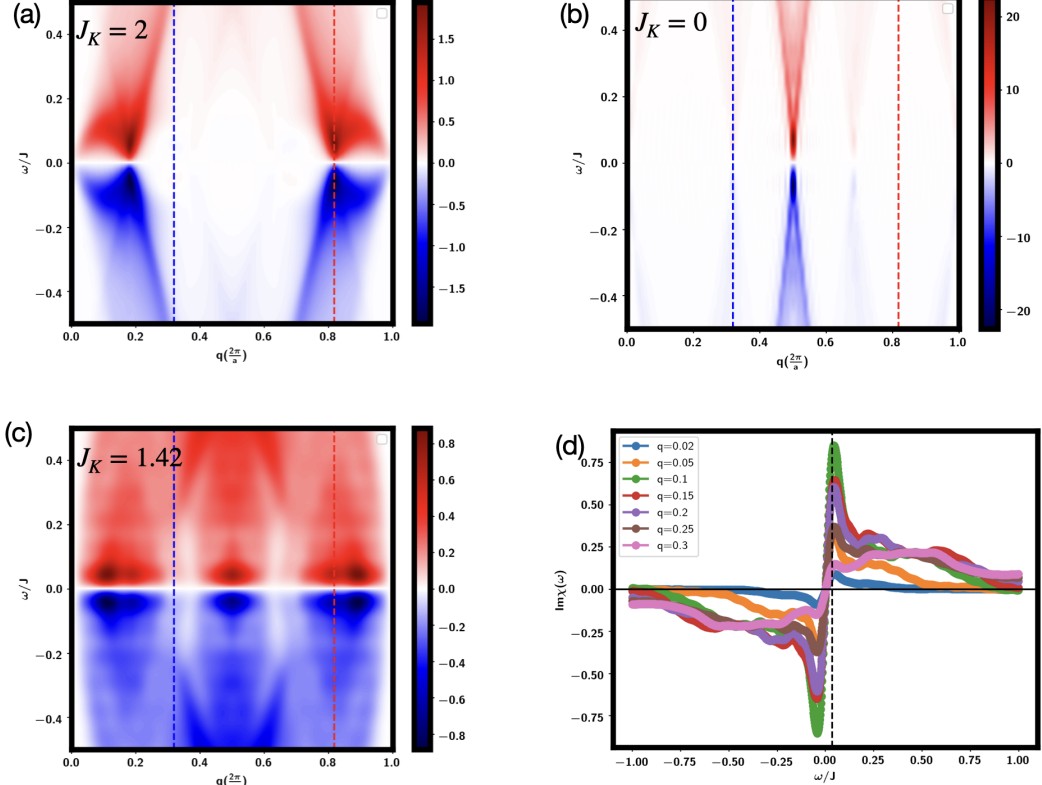

Figure 9: Dynamical spin structure factor $\mathrm{Im}\chi_{+-}(\omega, q)$ at $J_{cs} = 0$ and $x = \frac{7}{11}$. Here we use a system size of $L = 110$ in finite DMRG. (a) $\mathrm{Im}\chi_{+-}(\omega, q)$ for the LL phase at $J_K = 2$. There is a gapless mode at $q = 0$ and $2k_F = \frac{1+x}{2} \times 2\pi$. (b) LL* phase at $J_K = 0$. Note that the colour bar is significantly larger than other plots due to the large contribution from $\mathbf{Q} = \pi$ from the local spin moments. There is also a gapless mode at $q = 2k_F^* = \frac{x}{2} \times 2\pi$ corresponding to a small Fermi surface, but its spectral weight is smaller than that at $q = \pi$. (c) The unusual ultra-local critical behaviour at $J_K = 1.42$. (d) Line cuts along several momenta $q$ (in units of $\frac{2\pi}{a}$) at $J_K = 1.42$. The vertical dashed line is at $0.035J$. Below $0.035J$ the spectral weight vanishes as proportional to $\omega$, but this is due to numerical accuracy with a finite time evolution. In (a)(b)(c), the vertical red dashed line is at $2k_F = \frac{1+x}{2} \times 2\pi$, while the blue dashed line is at $2k_F^* = \frac{x}{2} \times 2\pi$. In the TEBD calculation, the total time is $T = 100$ with a step $\delta t = 0.15$ and the maximal bond dimension is $m = 500$. We use $\eta = 0.035$ for the damping term.

In contrast, at $J_K = 1.42$, there are gapless spin fluctuations in a range of momentum instead of just one single momentum point. From the line cut at several momenta (see Fig. 9(d)), we can see that the spectral weight grows when decreasing $\omega$ for a range of momentum until it reaches a cutoff energy scale (around $0.035J$) below which our calculation can not resolve. We note that the TEBD calculation is not quantitatively accurate, but qualitatively these results suggest that there is no dispersion within our numerical resolution.

As the dynamical spin structure factor $\mathrm{Im}\chi_S(\omega, q)$ is not accurate at the low energy limit, it is not clear whether the local criticality can survive down to zero energy limit or not. To understand the property at the zero energy limit, we need to rely on the ground state calculation. From the ground state calculation in finite DMRG (shown in Fig. 8) we already know that the spin correlation length is finite with $\xi_S \approx 10$ at $J_K = 1.42$. For a regular phase, we must also

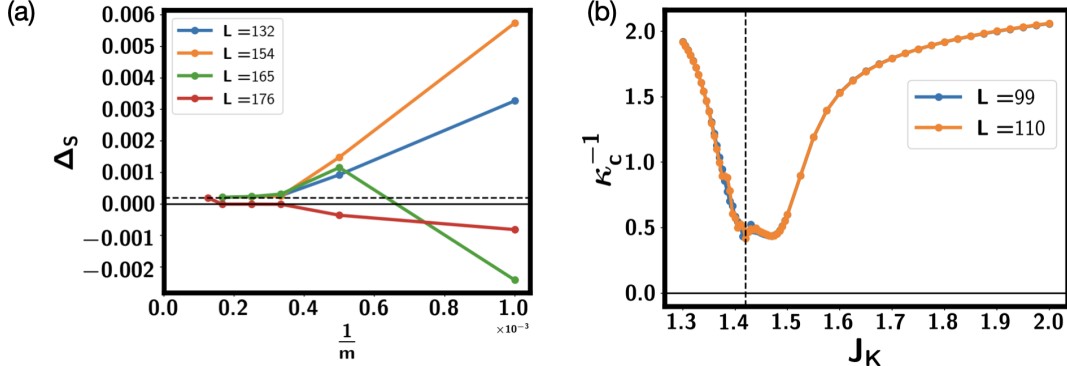

Figure 10: (a)Spin gap $\Delta_S$ from different system sizes at filling $x = \frac{7}{11}$, $J_K = 1.42$ and $J_{cs} = 0$. $\Delta_S$ is obtained as $E(S_z = 1) - E(S_z = 0)$. The bond dimension $m$ ranges from 1000 to 6000. For $L = 176$, the bond dimension is up to 8000. The dashed horizontal line indicates a gap of $2 \times 10^{-4}$. (b) Inverse charge compressibility $\kappa_c^{-1}$ with $J_K$ obtained from finite dmrg with bond dimension $m = 2000$. $\kappa_c = \frac{\partial n}{\partial \mu}$. For finite size $L$ with $N$ number of electron, we use the formula: $\kappa_c^{-1} = \frac{L}{4}(E(N+2)+E(N-2)-2E(N))$. At $J_K = 1.42$, $\kappa_c^{-1}$ remains finite, indicating that this is still a compressible phase. Actually, the compressibility is largest around $J_K = 1.42$.

have a finite spin gap $\Delta_S \propto \xi_S^{-1}$. We scale the spin gap at $J_K = 1.42$ with bond dimension up to $m = 6000$ ($m = 8000$ for $L = 176$) in Fig. 10(a). The conclusion is that there is an almost zero spin gap $\Delta_S \approx 2 \times 10^{-4}$. We conjecture that the gapless modes in a finite momentum region found in $\mathrm{Im}\chi_S(\omega, q)$ can survive down to this scale $\Delta_S \approx 2 \times 10^{-4}$. Note that $\Delta_S$ may still become truly zero if $J_K$ is fine-tuned to a critical point $J_K^c \approx 1.42$. Because the calculation is quite time-consuming, it is impossible for us to do a dense sampling around $J_K = 1.42$. Therefore it is still an open question whether the minimal spin gap is truly zero or not. However, even if there is a gap $\Delta_S \lesssim 2 \times 10^{-4}$ at true $J_K^c$, the ultra-local criticality behaviour still applies for the temperature scale above it. Given that almost any experimental measurement is likely performed well above this energy scale, we may conclude that ultra-local criticality exists for practical purposes.

Lastly, we comment on the density correlations. The inverse charge compressibility $\kappa_c^{-1}$ in Fig. 10(b) shows a dip around $J_K = 1.42$, indicating that this point is still a compressible phase with zero charge gap. Meanwhile in Fig. 7(d) and Fig. 8(d), we find that the correlation length in the density channel $\xi_N$ is also finite. It is then possible that there is also ultra-local critical behaviour in the density channel.

## 4.2 An intermediate weak ferromagnetic phase and ultra-local criticality

In the previous subsection, we find two spin-gapped domes for doping $x$ around $0.61 - 0.63$. Away from this narrow doping regime, we do not find another $LE_2$ phase. Instead, there is a very narrow but finite intermediate region which hosts a very weak ferromagnetic (FM) moment and also ultra-local criticality around the phase boundary.

In Fig. 11 we show that the inverse spin susceptibility $\chi_S^{-1} \propto \Delta_S L$ goes to basically zero in an intermediate region of $J_K$ at $x = \frac{4}{7}$ for both $J_{cs} = 0$ and $J_{cs} = 0.5J$. It also happens at a larger filling $x = \frac{21}{31}$ (see Fig. 11(c)(d)), suggesting that this is a quite generic phenomenon. In the following we focus on the parameter $x = \frac{21}{31}$ and $J_{cs} = 0.5J$. A vanishing $\Delta_S L$ in the

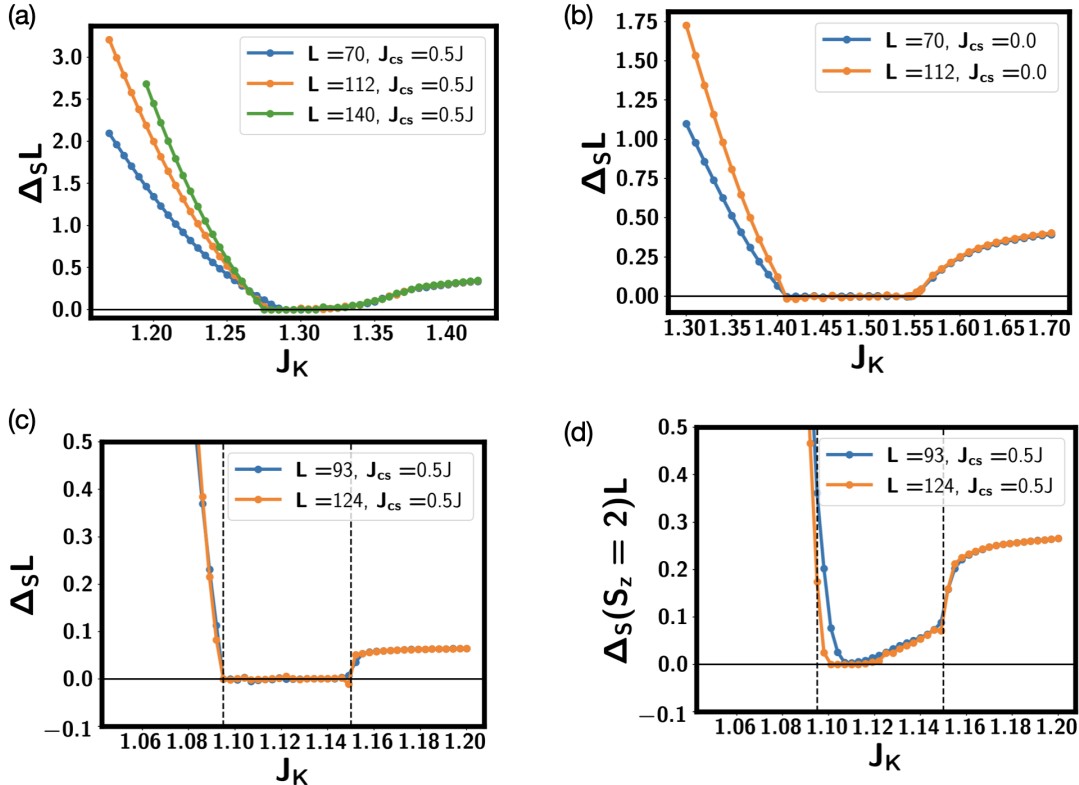

Figure 11: (a)$\Delta_S L$ with $J_K$ for $J_{cs} = 0.5J$, $x = \frac{4}{7}$ from finite DMRG for a few system sizes. $\Delta_S = E(S_z = 1) - E(S_z = 0)$ is the spin gap and $\Delta_S L$ is proportional to the inverse of the uniform static spin susceptibility. Here bond dimension is $m = 2000$. (b) $\Delta_S L$ at $x = \frac{4}{7}$ for $J_{cs} = 0$. Here bond dimension is $m = 2000$. (c) $\Delta_S L$ at $x = \frac{21}{31}$ and $J_{cs} = 0.5J$. (d) $\Delta_S(S_z = 2)L$ at $x = \frac{21}{31}$ and $J_{cs} = 0.5J$. $\Delta_S(S_z = 2) = E(S_z = 2) - E(S_z = 0)$. In (c)(d) the two dashed vertical lines label $J_K = 1.095$ and $J_K = 1.15$.

intermediate region indicates a divergence of the uniform spin susceptibility $\chi_s = \frac{\partial S_z}{\partial h}$ where $h$ is the Zeeman field. It usually signatures an FM phase. However, here we find that the FM moment is very small and only at the order of 1%. For an FM phase, we expect that $\Delta_S(S_z) = 0$ for $S_z < ML$, where $M$ is the ferromagnetic moment per site. In Fig. 11(d) we find that the gap of two spin flips becomes finite in this intermediate region except in a much smaller interval. Especially at the two boundaries $J_K = 1.095$ and $J_K = 1.15$ there is a finite $\Delta_S(S_z = 2)$, indicating that $M < \frac{2}{L}$ if there is an FM order.

In Fig. 12 we provide spin-spin correlation functions from $J_K = 1.09$ to $J_K = 1.14$ for $J_{cs} = 0.5J$ and $x = \frac{21}{31}$. They are obtained from finite DMRG, so boundary effects may matter here. One can see that the correlation function saturates to a small but finite value, which should be identified as $M^2$, where $M$ is the FM moment. We can see that around $J_K = 1.12 - 1.14$, the FM magnetic moment $M$ is at order $10^{-2}$. At $J_K = 1.09$, it is much smaller and at order $M \sim 10^{-4}$, which even decreases with the system size $L$. We note that a small FM moment of $M \sim 10^{-2}$ is roughly polarizing one spin in the entire system with $L \sim 100$. Hence it is even not clear whether this weak FM moment survives to the thermodynamic limit. We tend to conjecture that the weak FM moment is only a secondary effect in this region.

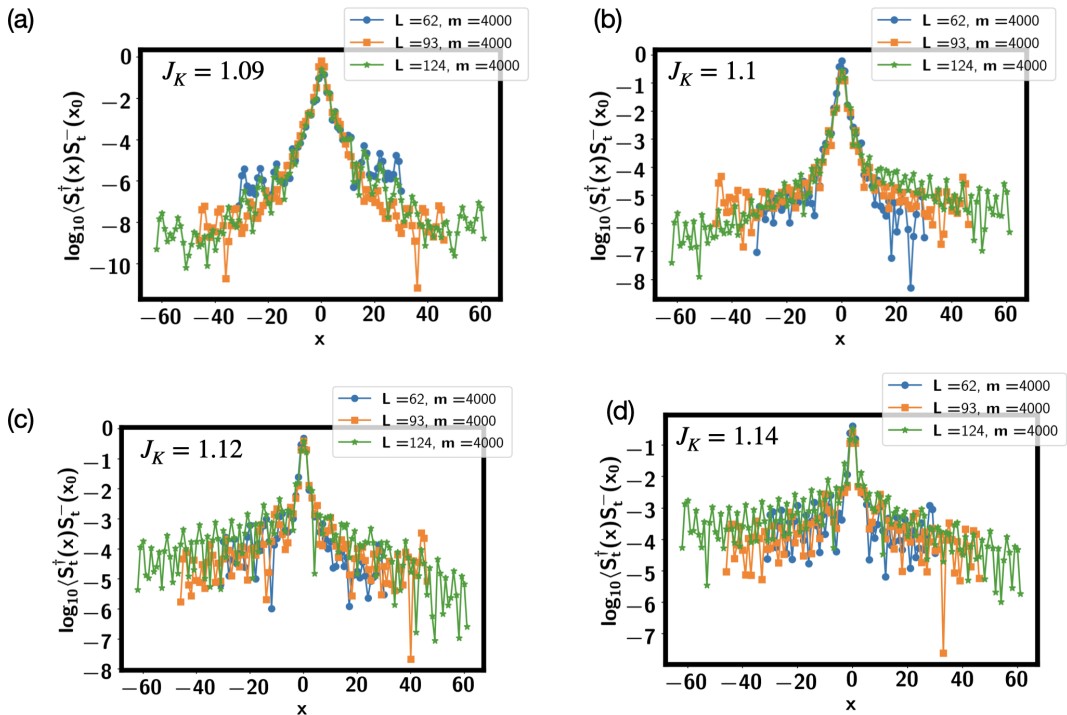

Figure 12: Spin-spin correlation functions from finite DMRG results at $J_{cs} = 0.5J$, $x = \frac{21}{31}$. (a) $J_K = 1.09$; (b) $J_K = 1.1$; (c) $J_K = 1.12$; (d) $J_K = 1.14$.

Despite the weak FM moment, we still discover ultra-local criticality behaviour in the dynamical spin structure factor at the two boundaries of this intermediate phase. In Fig. 13 we plot the imaginary dynamical spin susceptibility at the right boundary $J_K = 1.15$ and the left boundary $J_K = 1.1$ of the weak FM phase at $J_{cs} = 0.5J$ and $x = \frac{21}{31}$. At $J_K = 1.15$, one can see gapless spin modes in a range of momentum around $q = 0$. At $J_K = 1.1$, the gapless spin modes are mainly concentrated at $q = \pi$. From the line cuts at fixed momentum in Fig. 13(d) we can see that $\text{Im}\chi(\omega, q)$ in a range of $q$ around $q = \pi$ have constant spectral weight at intermediate energy and then grows at lower energy at $J_K = 1.1$. Within our numerical resolution, we can not see obvious dispersion, which suggests $z = \infty$ at least above the energy scale corresponding to our energy resolution (around $0.035J$). Spin fluctuations around $q = \pi$ should be mainly from the localized spin moments, suggesting that they are not fully Kondo screened at this parameter.

Inside the weak FM phase, we already know that there is a very small weak FM moment $M \sim 1\%$. However, we find the real part of the dynamical spin susceptibility is still dominated by $q = \pi$ instead of $q = 0$ as can be seen in Fig. 14. At $J_K = 1.08$, the system is still in a spin-gapped phase, one can see that the imaginary part of the dynamical spin susceptibility has spectral weights mainly around $q = \pi$ with the spin gap already very small. Correspondingly, the real part of the dynamical spin susceptibility is largest around $q = \pi$. The real susceptibility at $q = 0$ is very weak here. We can also approach the weak FM phase from large $J_K$. At $J_K = 1.15$ (see Fig. 14(d)) we find the real part of the dynamical spin susceptibility is large around $q = \pi$ and around $q = 2k_F = \frac{1+x}{2}$. The susceptibility at $q = 0$ is again quite small here. Then when we decrease $J_K$ to $J_K = 1.14$, there is some feature in $\text{Re}\chi_{+-}(\omega, q)$ around $q = 0$, but the region with largest susceptibility is still around $q = \pi$ (see Fig. 14(c)). All of these results suggest that the weak FM moment is likely only a secondary effect, not the main property of this region.

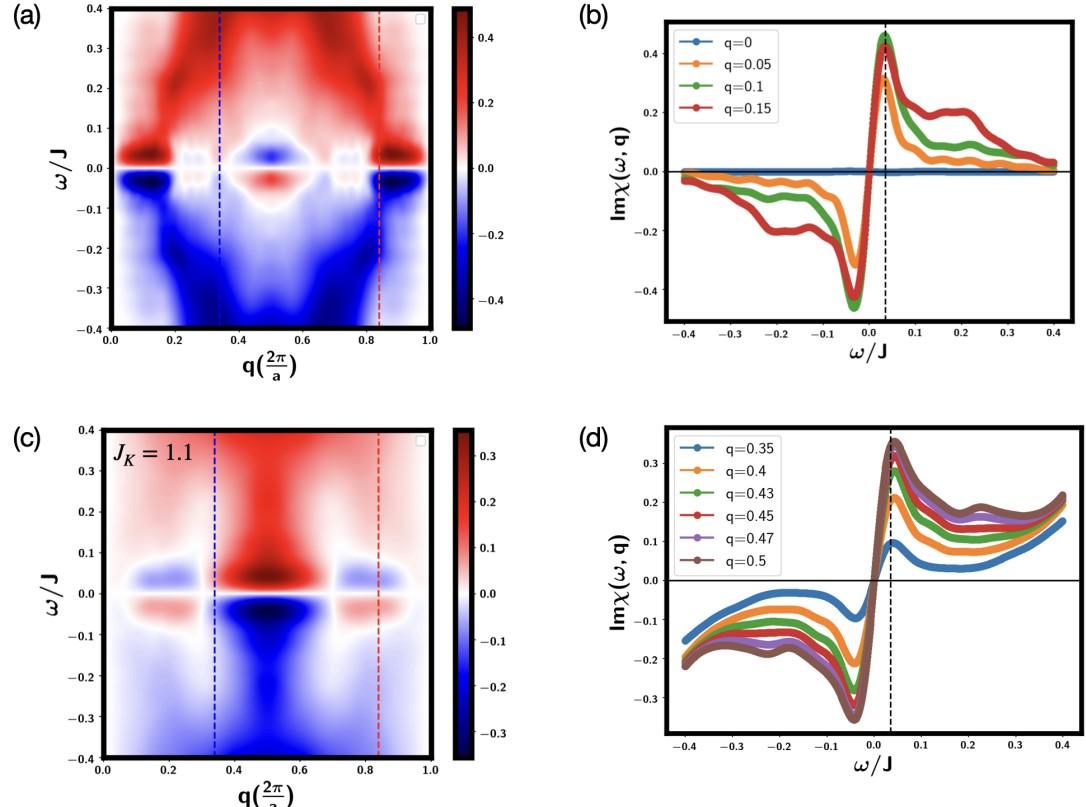

Figure 13: Dynamical spin susceptibility $\chi_{+-}(\omega, q)$ at the two boundaries of the weak FM phase at $J_{cs} = 0.5J$ and $x = \frac{21}{31}$. The system size is $L = 62$ in this calculation. (a) $\text{Im}\chi_{+-}(\omega, q)$ at $J_K = 1.15$. (b) Line cuts of $\text{Im}\chi_{+-}(\omega, q)$ at several $q$ at $J_K = 1.15$. (c) (d) are at $J_K = 1.1$. The calculation is done from the TEBD algorithm with total evolution time $T = 200$ with a step $\delta t = 0.1$. The bond dimension is $m = 2000$. We include a damping term $e^{-\eta t}$ with $\eta = 0.025$ when performing the Fourier transformation along the time direction. The dashed vertical line is at $\omega = 0.035J$. the momentum $q$ is in units of $2\pi/a$, where $a$ is the lattice constant.

Next, we try to offer a possible explanation for the weak FM order. In the Appendix E, we will show that the dynamical spin structure factors inside the weak FM phase (such as $J_K = 1.12$ and $J_K = 1.14$) have gapless spin fluctuations in a region around $q = 0$. Such many gapless fluctuations may couple to the boundary of the system and order at certain small momentum including $q = 0$. Our interpretation is that the spin modes get dispersion-less in a region around $q = 0$ and thus is very easy to be stuck in a profile with zero momentum or a small momentum. In real experiments with even weak disorder, we conjecture the system will develop a spin glass order. However, we note that the weak FM or spin glass order has only a very small moment and we should still expect ultra-local critical behaviour above a very small energy scale.

## 5 Discussion

Here we discuss the implications of the unusual behaviour we found at intermediate $J_K$ (phase boundary between LE and region I (or region II) of Fig. 2). One common property of the

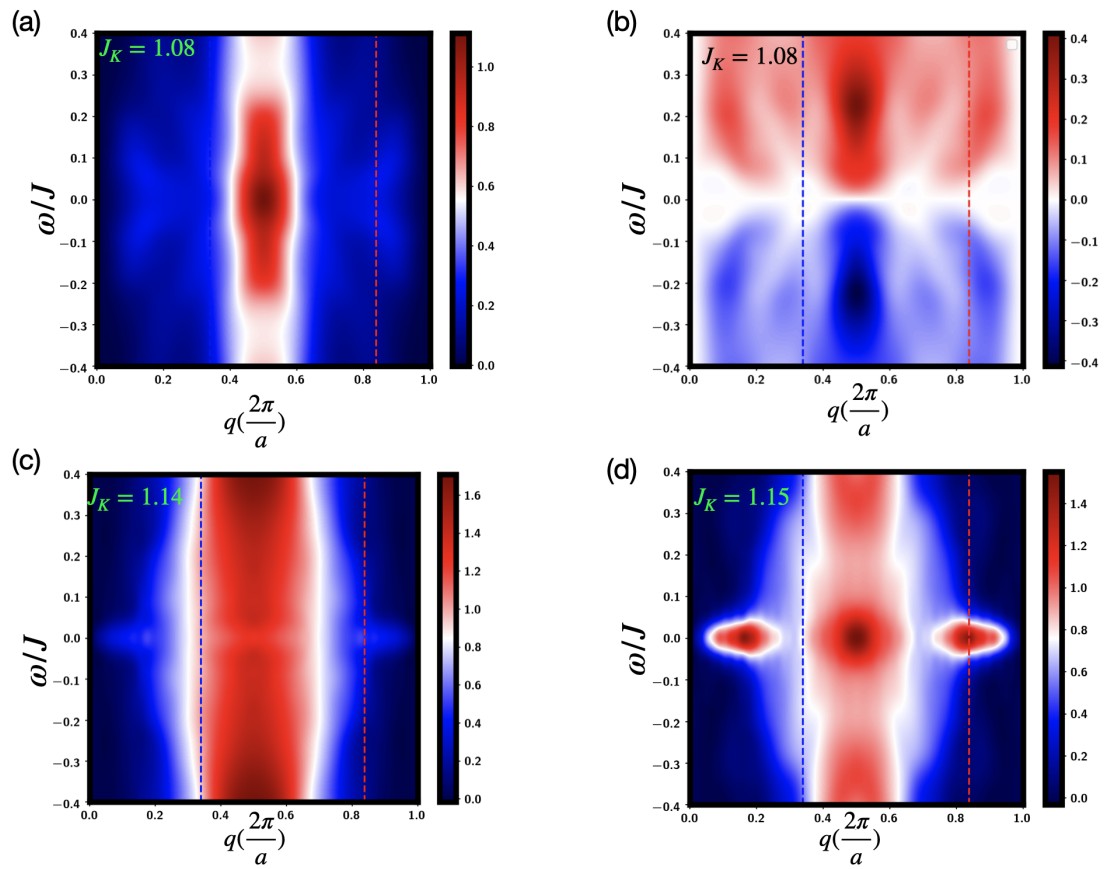

Figure 14: (a) $\mathrm{Re}\chi_{+-}(\omega,q)$ at $J_K = 1.08$. (b)$\mathrm{Im}\chi_{+-}(\omega,q)$ at $J_K = 1.08$. (c)$\mathrm{Re}\chi_{+-}(\omega,q)$ at $J_K = 1.14$. (d)$\mathrm{Re}\chi_{+-}(\omega,q)$ at $J_K = 1.15$. The parameters are the same as in Fig. 13.

intermediate narrow region is that the spin velocity $v_s$ apparently becomes small and even vanishes. So the question is how to understand a vanishing spin velocity or spatial correlations.

One of the ways in dealing with a Kondo lattice model is through the Abrikosov fermion theory:

$$\vec{S}_i = \frac{1}{2}f^\dagger_{i;\sigma}\vec{\sigma}_{\sigma\sigma'}f_{i;\sigma'}\,. \tag{2}$$

Then a Kondo screened phase at large $J_K$ is captured by a simple mean-field ansatz:

$$H_M = -b\sum_i c^\dagger_{i;\sigma}f_{i;\sigma}\,, \tag{3}$$

where $b \neq 0$ describes a Kondo-screened phase.

If we consider a model with $J = 0$ for the localized spin and also ignore the Ruderman–Kittel–Kasuya–Yosida(RKKY) interaction, then the $f$ band is perfectly flat. In this case, one expects $b \sim e^{-A\frac{t}{J_K}}$ and one can get a very small velocity in the $J_K \to 0$ limit. This is the usual heavy Fermion picture.

However, the model considered in this paper is different. We have a quite sizable $J = 0.5t$ between the localized spin moments. Therefore, in the above theory, there is a sizable velocity $v_f$ for the f band itself and one should not expect a vanishing velocity in the mean field picture. A simple way to describe the Kondo breakdown transition is to let $b$ vanish at a critical $J_K^c$ starting from the large $J_K$ phase [19]. However, in this picture, one does not expect $v_f$ (or

the bandwidth of the $f$ band) to vanish at the Kondo breakdown transition. Therefore, we still expect dispersion in spin fluctuations. This is in contrast to our discovery where we find dispersion-less gapless spin fluctuations in a range of momentum space within our energy resolution. For example, in Fig. 13(c) we can see a gapless spin fluctuation continuum in a range of $(\omega, q)$ space around $q = \pi$. In 1D we indeed expect that the localized spin moments contribute a gapless mode at $q = \pi$ in the Kondo breakdown phase, but it should have a strong dispersion proportional to $J$ (for example, see Fig. 9(b) for the result at $J_K = 0$). It is clear that the dispersion (or spatial correlation) of the localized spin moments also gets suppressed when approaching the critical regime. This is beyond the usual mean-field theory [19] where only the hybridization $c_\sigma^\dagger f_\sigma$ vanishes, while the bandwidth of the $f$ band is still proportional to $J$. The lesson we learned is that in the 'critical regime' between small $J_K$ and large $J_k$, the spatial correlation of local spin moments can get suppressed and then they fluctuate locally in real space. This is a property not captured by any theory we are aware of and thus offers a theoretical challenge. One can of course argue that this property may be special to this one-dimensional model and is irrelevant to higher dimensions. However, we note that similar features were observed in some neutron scattering experiments of higher dimensional heavy fermion material [17, 35]. This suggests that the 'ultra-local critical' phenomenon may be universal and relevant also for the higher dimensions.

## 6 Conclusion

In summary, we present the DMRG results of a one-dimensional Kondo lattice model. Through varying the Kondo coupling $J_K$, we studied how the Kondo breakdown phase evolves to the Kondo-screened Luttinger liquid phase. Around the intermediate regime, we discover signatures of ultra-local criticality with dynamical exponent $z = +\infty$ in the spin fluctuations. Similar phenomena have been reported in neutron scattering experiments of certain heavy fermion materials [17, 35]. Momentum-independent density fluctuations have also been observed at optimal doping of hole-doped cuprates [39]. However, the inevitable existence of disorder in real materials complicates the interpretations of these experimental results. Our numerical observations in a simple translation invariant model suggest that ultra-local criticality can arise intrinsically without the disorder. The fact that it shows up even in a 1D model may suggest that the phenomenon is quite universal around small to large Fermi surface transition and may be dimension independent. Theoretically, it was proposed that local criticality may be key to the solution of the mysterious strange metal phase [11, 34]. We plan to study the transport properties of this model in the near future to test this idea. Given the simplicity of a one-dimensional model, we hope future work on the current model will lead to progress on a better understanding of ultra-local criticality and strange metal.

## Acknowledgements

We thank Subir Sachdev for the discussions and comments on the manuscript. YHZ thanks Collin Broholm, T. Senthil and Mingru Yang for the discussions. AN thanks Pavel A. Volkov, Brenden Roberts and Alexei M. Tsvelik for the discussions.

**Funding information** YHZ was supported by the National Science Foundation under Grant No. DMR-2237031. AN was supported by the U.S. National Science Foundation grant No. DMR-2002850. The DMRG simulations were performed using the TeNPy Library(version 0.10.0) [38]. Part of the numerical simulation was carried out at the Advanced Research Com-

puting at Hopkins (ARCH) core facility (rockfish.jhu.edu), which is supported by the National Science Foundation (NSF) grant number OAC 1920103.

## A  Layer selective Mott localization and Kondo lattice model in bilayer optical lattice

In this Appendix, we propose to simulate a Kondo lattice model in bilayer optical lattice, as has been experimentally realized in Ref. [40]. One new requirement now is that we need to add a potential difference $\Delta$ between the two layers. The system is described by a bilayer Hubbard model:

$$
\begin{aligned}
H = \Delta \sum_i n_{i;1} - t \sum_{a=1,2} \sum_{\sigma=\uparrow,\downarrow} \sum_{\langle ij \rangle} (c^\dagger_{i;a\sigma} c_{j;a\sigma} + h.c.) \\
- t_{12} \sum_{a=1,2} \sum_{\sigma=\uparrow,\downarrow} \sum_{\langle ij \rangle} (c^\dagger_{i;1\sigma} c_{j;2\sigma} + c^\dagger_{i;2\sigma} c_{j;1\sigma} + h.c.) - t_\perp \sum_{a,\sigma} \sum_i (c^\dagger_{i;1\sigma} c_{i;2\sigma} + h.c.) \\
- \mu \sum_{a=1,2} \sum_i n_{i;a} + \frac{U}{2} \sum_a \sum_i n_{i;a}(n_{i;a} - 1) + U' \sum_i n_{i;1} n_{i;2} \,,
\end{aligned}
\tag{A.1}
$$

where $n_{i;a} = \sum_\sigma c^\dagger_{i;a\sigma} c_{i;a\sigma}$ is the density at site $i$ for layer $a = 1, 2$. $n_i = n_{i;1} + n_{i;2}$ is the total density at site $i$. We also define the average density $n = \frac{1}{N_s} \sum_i n_i$, where $N_s$ is the total number of sites in the system. Here $a = 1, 2$ labels the two layers and $t_\perp$ is the inter-layer vertical tunnelling. A non-zero $\Delta > 0$ is caused by a displacement field or a potential difference between the two layers. We will stay in the limit $U \gg t$ and $U \gg U'$. We assume $t_\perp, t < \Delta < U - U'$. At density $n = 1$, we have a Mott insulator with one particle at the layer 2. Then at density $n = 1 + x$ with $x \in (0, 1)$, the doped additional particle enters the layer 1 to reduce the on-site Hubbard $U$. In this case the layer 2 is always Mott localized and provides a spin 1/2 moment. The itinerant electron in the layer 1 is described by a $t$-$J$ model which then couples to the local moment of the layer 2 through a Kondo coupling. At low energy we can deal with an effective Kondo lattice model, with the same Hamiltonian as in Eq. 1 in the main text.

$$
\begin{aligned}
H = -tP \sum_{<i,j>,\sigma} (c^\dagger_{i,\sigma} c_{j,\sigma} + h.c.)P + J_c \sum_{\langle ij \rangle} \vec{S}^e_i \cdot \vec{S}^e_j + (V - \frac{1}{4}J_c) \sum_{\langle ij \rangle} n_i n_j \\
+ J \sum_{<i,j>} \vec{S}_i \cdot \vec{S}_j + J_K \sum_i \vec{S}^e_i \cdot \vec{S}_i + J_{cs} \sum_{\langle ij \rangle} \vec{S}^e_i \cdot \vec{S}_j + \vec{S}_i \cdot \vec{S}^e_j \,.
\end{aligned}
\tag{A.2}
$$

The couplings of the model are related to the optical lattice parameters in the following way: $J_c = J = \frac{4t^2}{U}$, $J_{cs} = 2\frac{t_{12}^2}{U-U'-\Delta} + 2\frac{t_{12}^2}{U+U'+\Delta}$ and $J_K = 2\frac{t_\perp^2}{U-U'-\Delta} + 2\frac{t_\perp^2}{U-U'+\Delta}$. This correspondence can be derived from the second order perturbation theory. Note, that when $t_{12} \neq 0$, there are also three-site correlated processes when an electron hops from layer 1 to layer 2, and then returns back to the different site. We have dropped those terms to make the analysis of the model simpler.

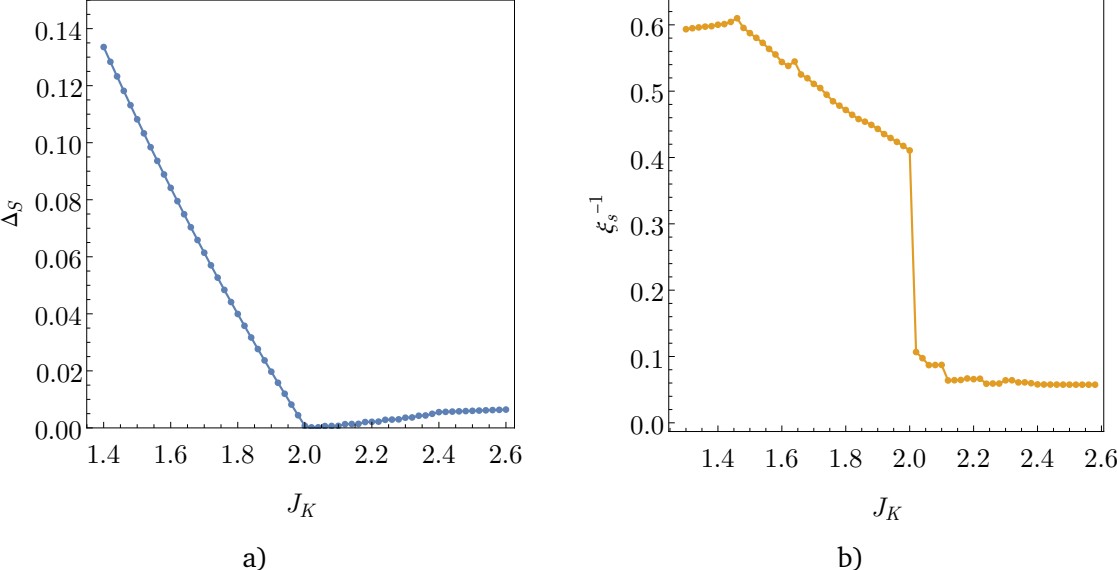

Figure 15: Spin gap and inverse correlation length at the transition between LE and intermediate I phase. The parameters are: $L = 113$, $x = 31/113 \approx 0.27$, $J_{cs} = 0.5J$, $V = 4J$, $m = 1000$.

## B Spin correlation length at the critical point near the intermediate region I

In this Appendix, we elaborate more on the transition between the LE phase and the intermediate region I phase. Fig. 15(a) shows the spin gap in the LE phase, gradually vanishing as we approach the transition point. Fig. 15(b) shows the inverse correlation length, extracted from the spin-spin correlation function. Right at the transition point $J_{Kc} \approx 2.0$ we again observe the signature of the ultra criticality: the spin gap is almost zero, while the inverse correlation length is finite $\xi_s^{-1} \approx 0.4$.

Assuming that transition between LE and I phase is of BKT type, the action in the massive phase is

$$S = \frac{1}{2\pi K c_s} \frac{1}{\beta \Omega} \sum_{k,n} \left( \omega_n^2 + c_s^2 k^2 + \Delta^2 \right) \phi_{k,n}^* \phi_{k,n} . \tag{B.1}$$

Based on this action, the most general relation between the correlation length and the gap is $\xi_s^{-1} = \Delta/c_s$. The natural way to obtain finite correlation length at zero spin gap is to have $c_s \to 0$. Note that $c_s$ is different from $v_s$, shown in Fig. 4(a) of the main text. While $v_s$ is the property of the gapless state, $c_s$ measures the dispersion of the excitation in the gapped phase. Therefore, $c_s = 0$ implies having a band of dispersionless excitations at energy $\Delta$ above the ground state. More detailed studies of this transition would be the purpose of our future work.

## C Convergence of spin correlation length at $J_K = 1.42, J_{cs} = 0, x = \frac{7}{11}$

We focus on $J_{cs} = 0$, $J_K = 1.42$ at $x = \frac{7}{11}$. In the main text, we show that the spin correlation length is finite while the spin gap is very small. Here we provide more evidence that the spin correlation length is indeed finite. In Fig. 16 we show the spin-spin correlation function with

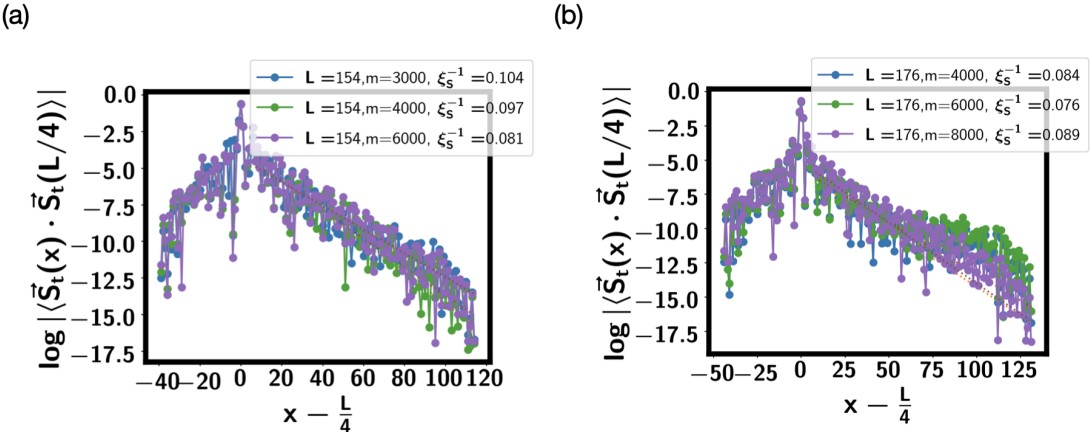

Figure 16: (a)(b) Spin-Spin correlation function with bond dimension at system size $L = 154$ and $L = 176$. Here we set the initial value $x_0 = L/4$. One can see that the correlation length $\xi_S$ becomes shorter when increasing the bond dimension for $L = 176$.

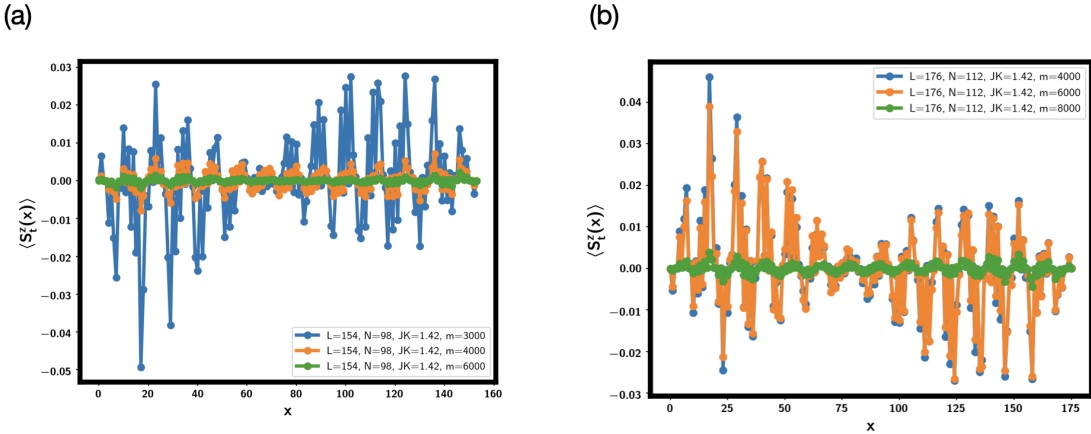

Figure 17: (a)(b) Total spin $\langle S_t^z(x) \rangle$ for system size $L = 154, 176$ at $J_K = 1.42$, $J_{cs} = 0$ and $x = \frac{7}{11}$.

bond dimension. At $L = 176$, we find that the correlation becomes more short-ranged when increasing the bond dimension, in agreement with a finite correlation length at the infinite bond dimension limit.

We also plot $\langle S_t^z(x) \rangle$ in Fig. 17. For a model with SU(2) spin rotation symmetry, we should expect $\langle S_t^z(x) \rangle = 0$ in the ground state. At the intermediate regime such as $J_K = 1.42$, $\langle S_t^z(x) \rangle$ vanishes quite slowly with the bond dimension. We need to use the bond dimension $m = 8000$ for $L = 176$.

Lastly in Fig. 18 we show that the same re-entrance of spin-gapped phase also exists at a different but close filling $x = \frac{19}{31}$ with $J_{cs} = 0$.

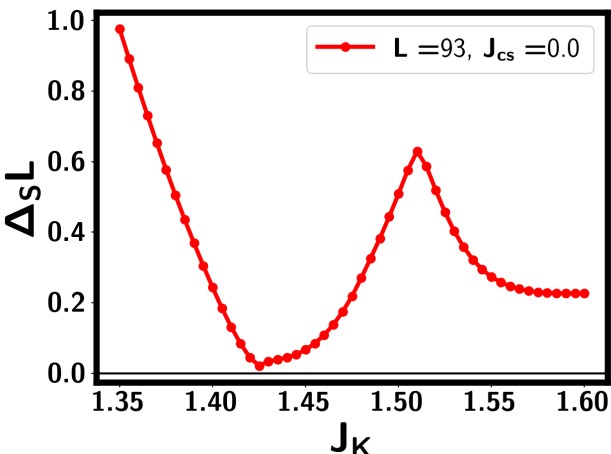

Figure 18: Re-entrance of spin gapped phase at a different filling $x = \frac{19}{31}$.

# D TEBD calculation of the dynamical spin susceptibility

We want to calculate the dynamical spin susceptibility:

$$\chi_{ij}(t - t') = i\langle[\vec{S}_i(t), \vec{S}_j(t')]\rangle\theta(t - t').\tag{D.1}$$

We can decompose $\chi_{ij}$ to be:

$$\chi_{ij}(t - t') = \chi_{ij}^{zz}(t - t') + \chi_{ij}^{yy}(t - t') + \chi_{ij}^{zz}(t - t').\tag{D.2}$$

We define:

$$C_{ab;ij}(t) = \langle S_i^a(t)S_j^b(0)\rangle - \langle S_i^a\rangle\langle S_j^b\rangle.\tag{D.3}$$

We will mainly focus on calculation $\chi_{ij}^{+-}(t - t')$. We have the following identity:

$$\chi_{xx;ij}(t) + \chi_{yy;ij}(t) = i\theta(t)(\frac{1}{2}(\chi_{+-;ij}(t) + \chi_{-+;ij}(t))),\tag{D.4}$$

where

$$\chi_{+-;ij}(t) = i\theta(t)(C_{+-;ij}(t) - C_{+-;ji}(-t)),\tag{D.5}$$

$$\chi_{-+;ij}(t) = i\theta(t)(C_{-+;ij}(t) - C_{-+;ji}(-t)).\tag{D.6}$$

We have the equation:

$$C_{+-;ij}^*(t) = C_{+-;ji}(-t).\tag{D.7}$$

Therefore

$$\chi_{+-;ij}(\omega) = i\int_0^\infty (C_{+-;ij}(t) - C_{+-;ij}^*(t))e^{i\omega t},\tag{D.8}$$

$$\chi_{-+;ij}(\omega) = i\int_0^\infty (C_{-+;ij}(t) - C_{-+;ij}^*(t))e^{i\omega t}.\tag{D.9}$$

In practice our evolution is limited to a finite time $T$, we use the following formula:

$$\chi_{-+;ij}(\omega) = i\int_0^T (C_{-+;ij}(t) - C_{-+;ij}^*(t))e^{i\omega t}e^{-\eta t},\tag{D.10}$$

where $\eta$ is a damping term.

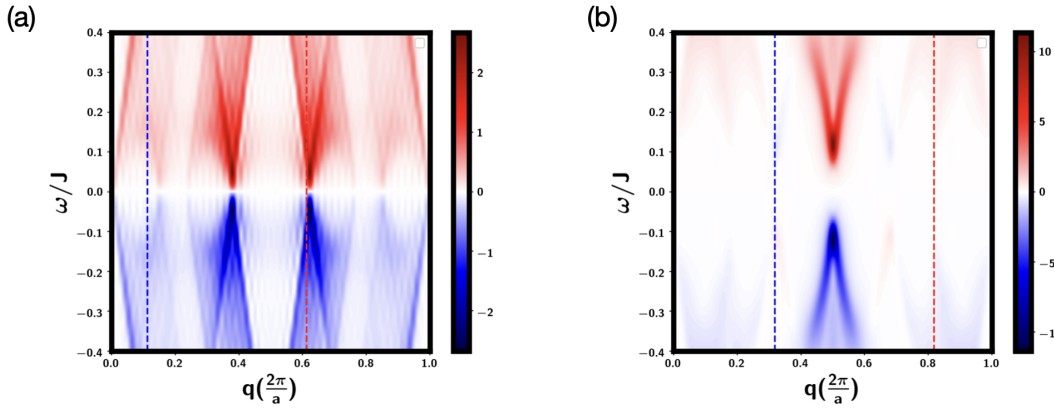

Figure 19: Im$\chi_{+-}(\omega, q)$. (a) $J_K = 3$, $J_{cs} = 0.5J$ and $x = \frac{21}{93}$ with $L = 93$. This is a Luttinger phase. The red and blue dashed lines are at $q = 2k_F = \frac{1+x}{2} \times 2\pi$ and $q = 2k_F^* = \frac{x}{2} \times 2k_F$ respectively. (b) $J_K = 1$, $J_{cs} = 0$, $x = \frac{7}{11}$ with $L = 110$. This is in the spin-gapped Luther-Emery phase. In both calculations, we use bond dimension $m = 500$, total time $T = 100$ and a step $\delta t = 0.15$. We use $\eta = 0.025$ in performing the Fourier transformation along the time direction.

For our model, the above two quantities are the same. Therefore we focus on $\chi_{+-}$. By doing Fourier transformation also in real space, we get:

$$\chi_{+-}(\omega, q) = \sum_i \chi_{+-;i,j=L/2}(\omega) \cos(q(i - L/2)). \tag{D.11}$$

Here we use $\cos(qx)$ instead of $e^{iqx}$ by assuming the inversion symmetry respect to $x = L/2$. This can remove the artificial inversion breaking from numerical inaccuracy.

In our calculation, we use the TEBD algorithm with $T = 200$ (assuming the hopping $t = 1$) with a step $\delta t = 0.1$. The largest bond dimension is $m = 500 - 2000$. When doing the Fourier transformation, we typically use $\eta = 0.035$. Note that without $\eta$ we will find oscillations in $\chi(\omega)$ because of the finite time $T$.

To benchmark the calculation, we first try two points deep inside the LL and the spin-gapped LE phase, shown in Fig. 19. Here we only use bond dimension $m = 500$, but one can see the results are already quite reasonable. In (a) we find gapless mode at $q = 2k_F$ and $q = 0$, typical behaviour of a Luttinger liquid phase. There are also features at higher harmonics. In (b) we find a clear spin gap in the LE phase. These results demonstrate the validity of the TEBD calculation.

To check that the TEBD results converge with the bond dimension even in the intermediate regime with ultra-local criticality, we plot Im$\chi_{+-}(\omega, q)$ at the left boundary of the weak FM phase at $J_{cs} = 0.5$, $J_K = 1.1$, $x = \frac{42}{62}$ in Fig. 20. We can see that the results are qualitatively the same for bond dimensions $m = 1000$ and $m = 2000$. Both show dispersionless gapless spin fluctuations around $q = \pi$.

## E  The weak FM phase

In this section we add more discussions on the weak FM phase in the region II.

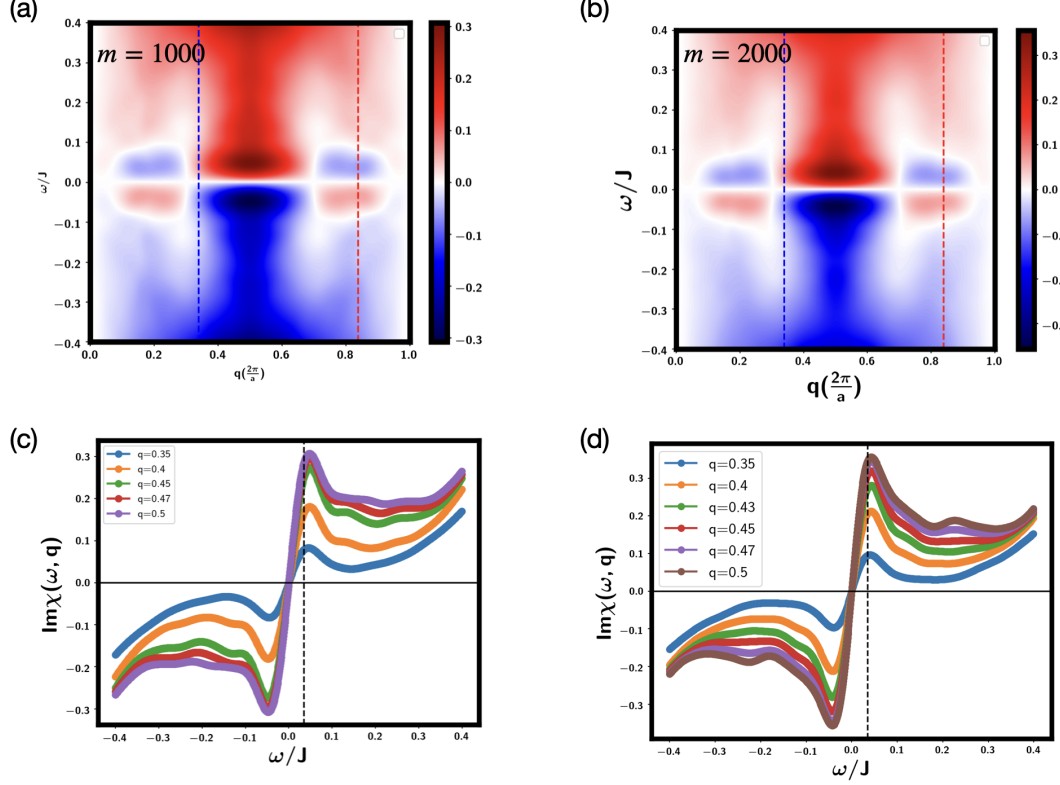

Figure 20: Change of $\text{Im}\chi_{+-}(\omega, q)$ with the bond dimension. Here $J_{cs} = 0.5$, $J_K = 1.1$, $L = 62$ and $x = \frac{42}{62}$. (a)(c) bond dimension $m = 1000$. (b)(d) bond dimension $m = 2000$.

## E.1  More results at $J_{cs} = 0.5J$, $x = \frac{21}{31}$

Here in Fig. 21 and Fig. 22 we show more data to supplement the discussions in the main text on the weak FM regime for the parameter $J_{cs} = 0.5J$, $V = 0$ at the filling $x = \frac{21}{31}$. These results are again obtained from the TEBD calculation with system size $L = 62$. From the $\text{Im}\chi_{+-}(\omega, q)$ at $J_k = 1.12, 1.14$ inside the weak FM phase, we can see gapless spin fluctuations in a region around $q = 0$.

## E.2  Another filling

We also provide the results from a different filling $x = \frac{4}{7}$ for the weak FM region. From infinite DMRG in Fig. 23, we can see that there is a $c = 2$ region (with spin gap) in $J_K \in [1.295, 1.305]$ and a $c = 3$ region in $J_K \in [1.32, 1.38]$. Compared to the finite DMRG in the main text (Fig. 11(a)), $\Delta_S L$ approaches zero in the whole $c = 2$ region and the left part of the $c = 3$ region. Between $J_K = 1.31$ and $J_K = 1.32$, we again see that the entanglement entropy $S$ grows with $\log m$ while the correlation length $\xi_N$ saturates. In the region $J_K \in [1.295, 1.305]$, there is a quite short correlation length in the spin channel, as shown in Fig. 24(a). This is again at odds with the vanishing $\Delta_S L$ from our finite DMRG calculation in Fig. 11(a). In this case, the charge correlation length is infinite and we have a regular central charge $c = 2$, presumably from two charge modes. However, the spin modes are not simply gapped given that $\Delta_S L$ goes to zero even faster than any power of $1/L$. We conjecture that in the spin channel there is still ultra-local criticality. When $J_K > 1.31$, we find that $|\langle S^\dagger(x)S^-(0)\rangle|$ seems to saturate to a finite value in the large $x$ limit, indicating a very small FM moment.

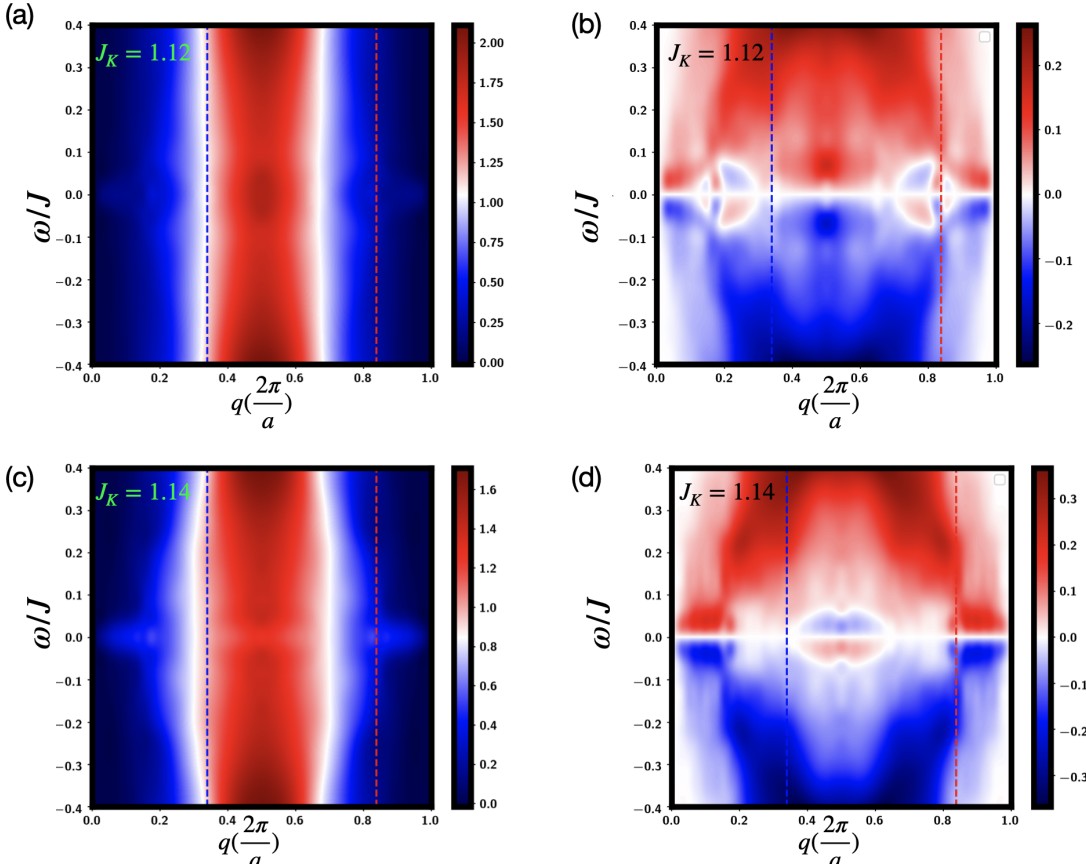

Figure 21: Dynamical spin susceptibility $\chi_{+-}(\omega, q)$ at $J_K = 1.14$ and $J_K = 1.12$. (a)(c) Re$\chi_{+-}(\omega, q)$. (b)(d) Im$\chi_{+-}(\omega, q)$. One can see gapless spectral weights in a region around $q = 0$.

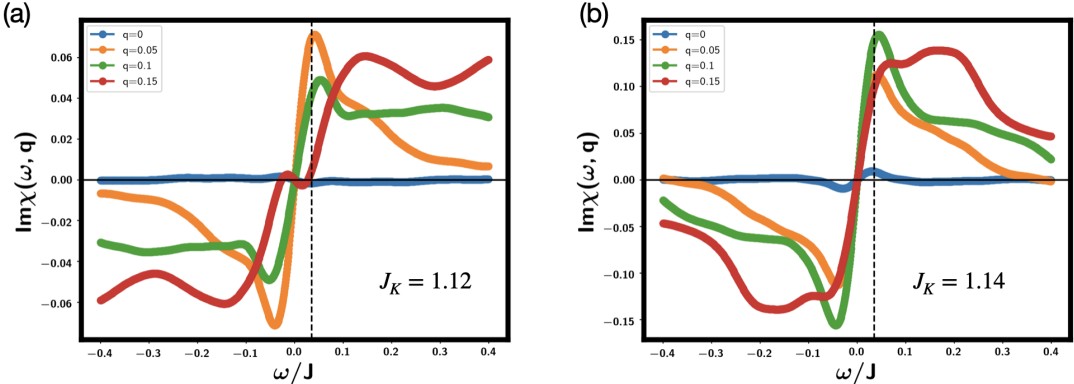

Figure 22: Line cut at fixed $q$ of Im$\chi_{+-}(\omega, q)$ inside the weak FM phase at $J_K = 1.12$ and $J_K = 1.14$.

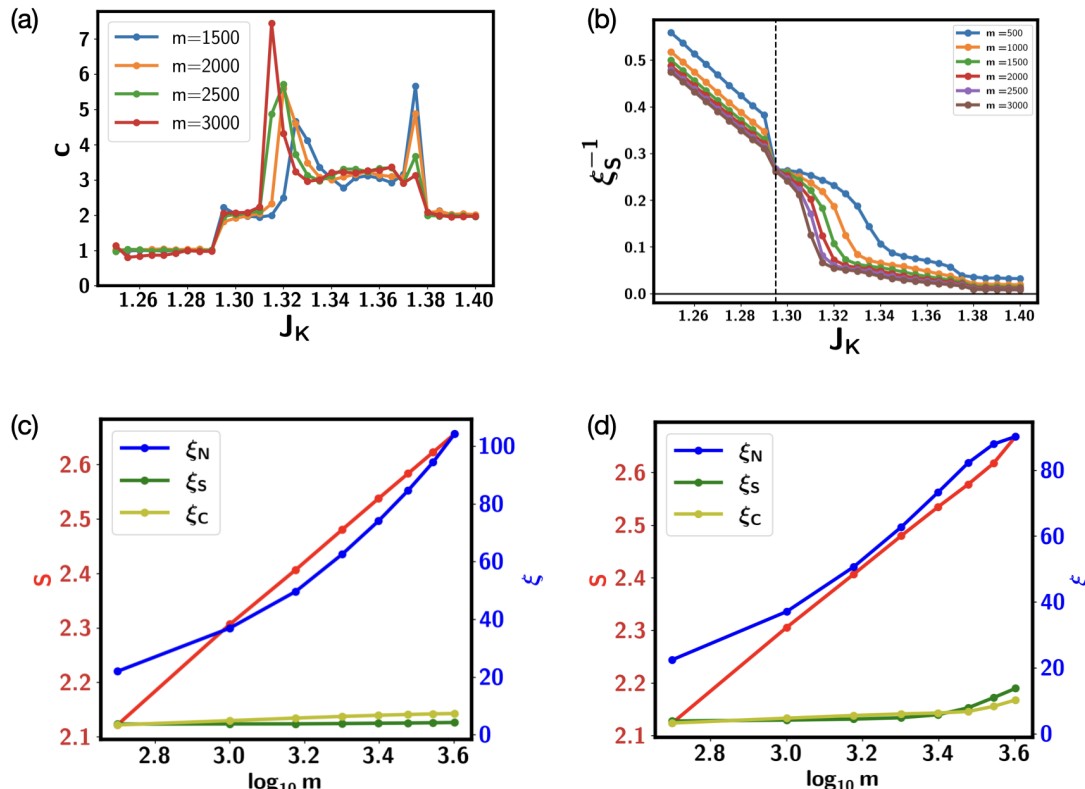

Figure 23: Infinite DMRG results at $J_{cs} = 0.5J$, $x = \frac{4}{7}$. We use unit cell size 22. (a) Central charge as a function of $J_K$. (b) The inverse of the spin correlation length $\xi_S^{-1}$ as a function of $J_K$. $\xi_S$ is obtained from the transfer matrix technique in the sector $(Q, S_z) = (0, 1)$. (c) Scaling of the entanglement entropy and correlation lengths with bond dimension $m$ at $J_K = 1.3$. $\xi_N$, $\xi_S$, $\xi_C$ correspond to density, spin and single electron operators, obtained from the sector $(Q, S_z) = (0, 0), (0, 1), (1, \frac{1}{2})$ respectively. (d) Scaling of the entanglement entropy and correlation lengths with the bond dimension $m$ at $J_K = 1.31$.

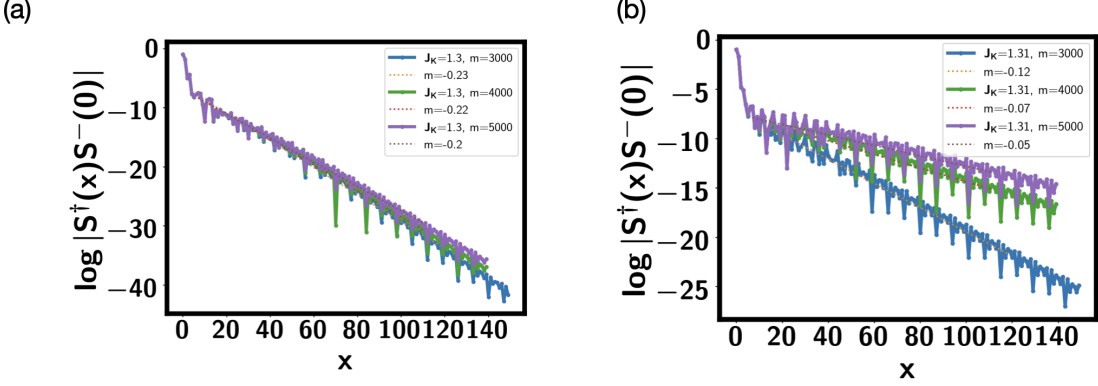

Figure 24: Log-x plot of the spin spin correlation function $\langle S^\dagger(x)S^-(0) \rangle$ from infintie DMRG at $x = \frac{4}{7}$, $J_{cs} = 0.5J$. (a) $J_K = 1.3$. (b) $J_K = 1.31$.

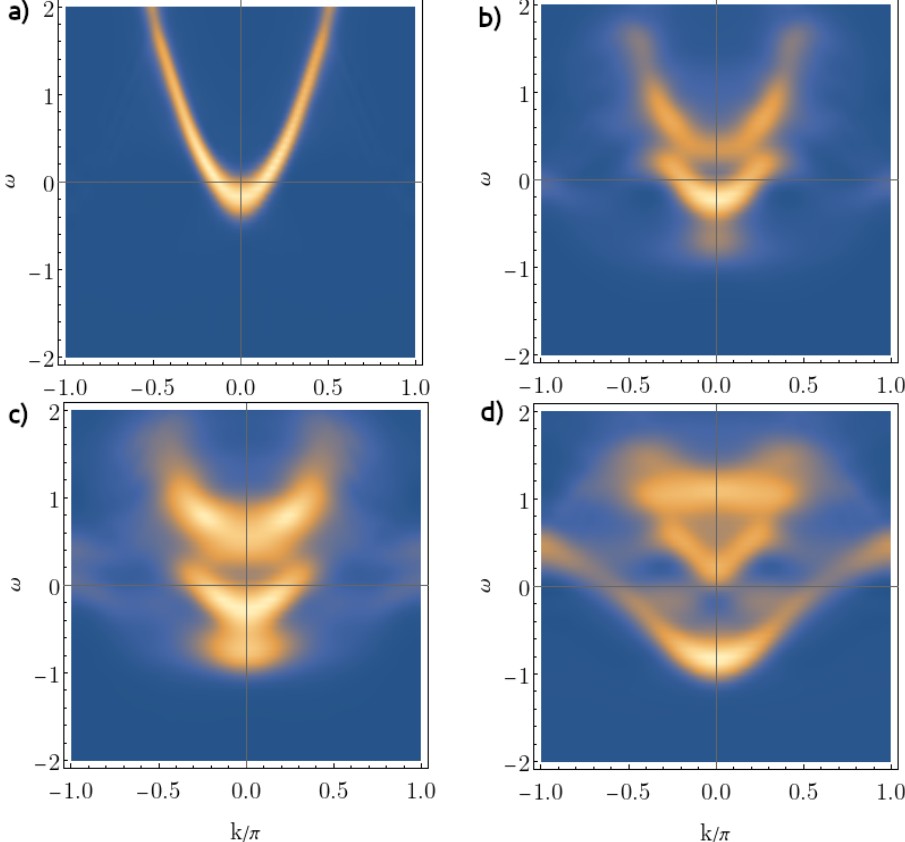

Figure 25: TEBD calculation of electron spectral function for Kondo-Heisenberg model. Figures a)-d) correspond to the Kondo coupling $J_K = 1, 2.3, 2.6, 4$ correspondingly, while other parameters are $J = 0.5, t = 1$. The length of the chain was $L = 80$ and the number of electrons $n_e = 20$, while the bond dimension $m = 500$ with the time evolution $t = 12$ and additional linear prediction interpolation.

## F  TEBD calculation of the single electron spectral density

In this appendix, we demonstrate the existence of the small and large Fermi momenta by calculating the electron spectral function. The spectral function is defined as $A_k(\omega) = -Im G_k^R(\omega)$ and $G_k^R(t) = \langle \{c_k(t), c_k^\dagger(0)\} \rangle$. The calculations were done for the Kondo-Heisenberg model to ensure that the Luther-Emery phase at $J_K \to 0$ coincides with the free electrons phase.

$$H = -t \sum_{<i,j>,\sigma} (c_{i,\sigma}^\dagger c_{j,\sigma} + h.c) + J_K \sum_i \vec{S}_i^e \vec{S}_i + J \sum_i \vec{S}_i \vec{S}_j. \tag{F.1}$$

We studied the model at $x = 0.25$ and observed the same phases as in Figure 2, with phase I being in the region $J_K \approx (2.5, 2.8)$. Fig. 25(a) shows the small Kondo coupling regime with the spectral function simply matching the free electron band with small Fermi momentum $2k_F = x/2$. The spin gap should also be present but it is too small to be distinguishable. At larger $J_K$ the layer of spins starts to interfere and the band dispersion is modified, see Fig. 25(b). A similar band reconstruction happens if we implement a mean-field theory. Fig. 25(c) shows the dispersion in phase I which has a $c = 3$ central charge and a split in Fermi momentum. We do not clearly see two Fermi momenta, but the bands are strongly reconstructed and the accuracy is not enough to make a definite conclusion. Finally, at large Kondo coupling the system is in LL phase with a large Fermi momentum $2k_F = (1 + x)/2$, see Fig. 25(d).

To obtain better frequency resolution we used a linear prediction algorithm, see [41]. We extrapolated to times 3 times larger than the initial computation. The energies also need to be shifted by the chemical potential. We computed it separately by using the formula $\mu_c = (E(n_e + 2) - E(n_e - 2))/4$.

## G  Mean-field theory

In this appendix, we show how some of the observations in the main part of the paper can be explained using simple mean-field analysis. As in the previous appendix, we start with the simpler Kondo-Heisenberg Hamiltonian

$$H = -t \sum_{<i,j>,\sigma} c^\dagger_{i,\sigma} c_{j,\sigma} + \sum_{<i,j>} \left( J \vec{S}_i \vec{S}_j + J_{cs} \vec{S}^e_i \vec{S}_j \right) + J_K \sum_i \vec{S}^e_i \vec{S}_i \,. \tag{G.1}$$

We fractionalize spin $\vec{S} = 1/2 f^\dagger_\alpha \sigma_{\alpha\beta} f_\beta$ and use Pauli identities to obtain

$$H = -2t \cos(ka) c^\dagger_k c_k - \frac{J}{2} f^\dagger_{i\alpha} f_{j\alpha} f^\dagger_{j\beta} f_{i\beta} - \frac{J_{cs}}{2} f^\dagger_{i\alpha} c_{j\alpha} c^\dagger_{j\beta} f_{i\beta} - \frac{J_K}{2} f^\dagger_{i\alpha} c_{i\alpha} c^\dagger_{i\beta} f_{i\beta} \,. \tag{G.2}$$

After taking a large $M$ limit (where $M$ is a number of spin indices), we arrive at the following saddle-point equations:

$$
\begin{aligned}
P &= -J_K \langle f^\dagger_i c_i \rangle = -\frac{J_K}{V} \sum_k \langle f^\dagger_k c_k \rangle \,, \\
Q &= -J \langle f^\dagger_i f_{i+1} \rangle = -\frac{J}{V} \sum_k \cos(ka) \langle f^\dagger_k f_k \rangle \,, \\
PR &= -J_{cs} \langle f^\dagger_i c_{i+1} \rangle = -\frac{J_{cs}}{V} \sum_k \cos(ka) \langle f^\dagger_k c_k \rangle \,, \\
\frac{1}{2} &= \frac{1}{V} \sum_k \langle f^\dagger_k f_k \rangle \,.
\end{aligned}
\tag{G.3}
$$

The corresponding free Hamiltonian is

$$
\begin{aligned}
H = &(-2t \cos(ka) - \mu) c^\dagger_k c_k + (2Q \cos(ka) + \lambda) f^\dagger_k f_k \\
&+ P(1 + 2R \cos(ka)) c^\dagger_k f_k + P(1 + 2R \cos(ka)) f^\dagger_k c_k \,.
\end{aligned}
\tag{G.4}
$$

We take $x = 2\rho_c = 0.7$, $J = 0.5$, $J_{cs} = 0.25$ and $t = 1$, which is close to the parameters in the paper, and study the phase diagram as a function of $J_K$. This requires solving Eq. G.3 self-consistently. At small $J_K < 1.4$ there is only a trivial solution for the mean-field $P = 0$ which corresponds to an LL$^*$ phase with a small Fermi surface $k_F = \pi x/2$. For the intermediate $J_K \in [1.4, 2]$ there is a nontrivial solution which corresponds to a nontrivial hybridized Fermi-surfaces, see Fig. 26(a). There is another solution with two Fermi surfaces, see Fig. 26(b), which proves to be unstable after a comparison of Free energies. Finally, at large $J_K > 2$, there is a nontrivial solution with a large Fermi surface $k_F = \pi(1 + x)/2$, which correspond to an LL phase, see Fig. 26(c).

The point of the analysis above is to show that a simple mean-field model is able to capture certain properties of the phase diagram, such as a transition from an LL$^*$ to an LL phase through the intermediate phase with two FS. Though certain features, such as the spin gap, are missing from the above analysis, the rough boundaries of the phases match our DMRG predictions in Fig. 2.

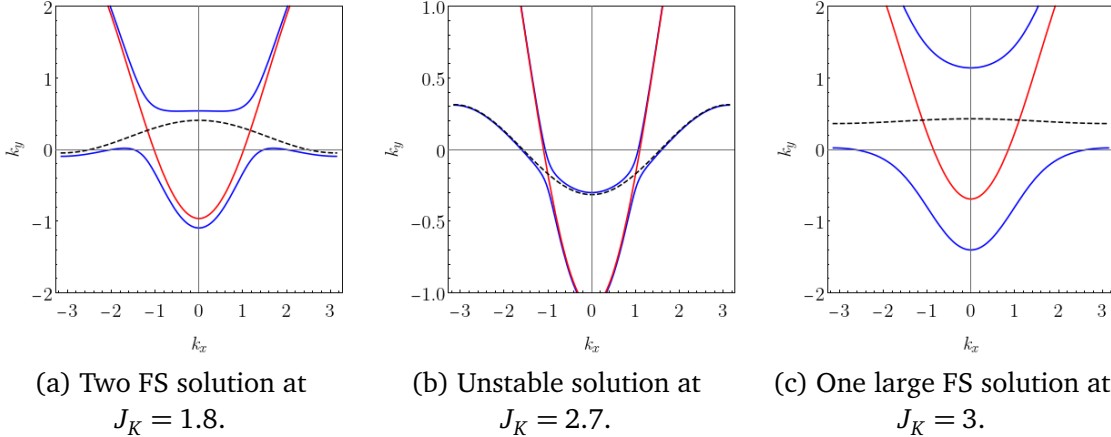

(a) Two FS solution at
$J_K = 1.8$.

(b) Unstable solution at
$J_K = 2.7$.

(c) One large FS solution at
$J_K = 3$.

Figure 26: The energy bands of the Hamiltonian in Eq. G.4 at different $J_K$. The red line is a dispersion of a $c$ particle, the dashed black line is a dispersion of an $f$ particle, and the blue lines are full hybridized dispersions, given by eigenvalues of $H$.

Now we address another question, stated in Section 4 of the paper, namely the existence of the phase with zero gap and finite correlation length. To evaluate the correlation length, we computed the density-density response of the Hamiltonian in Eq. G.4. Equal time density-density correlation in Fourier space is:

$$\langle N(q)N(-q)\rangle = \sum_{k,\omega_n,q_n} G_c(k+q,\omega_n+q_n)G_c(k,\omega_n). \tag{G.5}$$

For the Kondo-Heisenberg model, the Green function $G_c$ is

$$G_c = \frac{1}{i\omega_n - E_+}\frac{E_+ - e_f}{E_+ - E_-} + \frac{1}{i\omega_n - E_-}\frac{E_- - e_f}{E_- - E_+} = \frac{u_+^2}{i\omega_n - E_+} + \frac{u_-^2}{i\omega_n - E_-}, \tag{G.6}$$

where $E_+$ and $E_-$ are the eigenvalues of the Hamiltonian $H$. The full density-density correlation functions are

$$\langle N(q)N(-q)\rangle = \sum_k (n_F(E_{+q}) - n_F(E_+)n_B(E_{+q} - E_+)u_+^2 u_{+q}^2 + (n_F(E_{-q})$$
$$- n_F(E_-)n_B(E_{-q} - E_-)u_-^2 u_{-q}^2 + +(n_F(E_{+q}) - n_F(E_-)n_B(E_{+q} - E_-)u_-^2 u_{+q}^2$$
$$+ (n_F(E_{-q}) - n_F(E_+)n_B(E_{-q} - E_+)u_+^2 u_{-q}^2, \tag{G.7}$$

where each term correspond to scattering from the band $E_i$ to the band $E_j$, $i,j = +,-$. The leading contribution will be given by the scattering in the lower band $E_- \to E_-$, since this band hosts zero-energy excitations. We computed the density-density response for the simple mean-field model in Fig. 27(a) at two temperatures. At small temperatures, we observed a typical power law decay with an infinite correlation length, see Fig. 27(b). However, at finite temperatures comparable to the dispersion of the $f$ electron, the density-density correlation demonstrated an exponential decay, with finite correlation length and zero spin gap. Though our DMRG calculations are done at zero temperature, there is always a finite error in energy calculation $\delta E_s$. Therefore, we can successfully explain the existence of a finite correlation length and zero spin gap by assuming that the dispersion of the $f$ electron is even smaller $Q \ll \delta E_s$. $\delta E_s$ is very small in our calculation, and the $f$ fermion band should be almost flat, which is quite a surprise for a finite $J$.

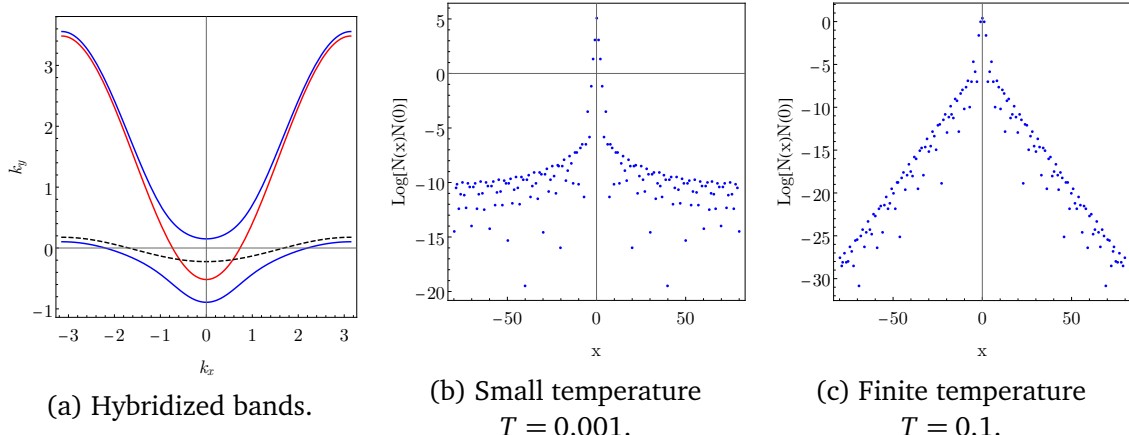

(a) Hybridized bands.     (b) Small temperature $T = 0.001$.     (c) Finite temperature $T = 0.1$.

Figure 27: (a) The hybridized bands for parameters $x = 0.4$, $Q = -0.1$, $P = 0.5$, $R = 0$ and $t = 1$. (b) Density-Density response at small temperatures in logarithmic scale shows power-law decay with $\xi^{-1} = 0$. (c) Density-density response at finite temperatures in logarithmic scale shows exponential decay with finite correlation length.

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
