# Peer review of "Numerical signatures of ultra-local criticality in a one dimensional Kondo lattice model"

_SciPost Physics, doi:SciPost Phys. 17, 034 (2024)_

## Round 1 · Referee Report · Anonymous (Referee 2) · 2024-1-2

Strengths

  1. The paper has carried out a thorough DMRG analysis of a 1D Kondo lattice with onsite and intersite Kondo exchange terms.

  2. The results indicate the formation of an unconventional critical point at intermediate coupling with gapless spin excitations co-existing with a finite correlation length, and an entanglement entropy that can not be fitted with a simple conformal charge, suggesting some kind of local criticality that can not be simply described by conformal field theory.

Weaknesses

Weaknesses: 1. The Kondo lattice model studied omits co-tunneling three-site terms which would appear in a Schrieffer Wolff transformation of the original two layer Anderson model: as such, the model studied is not obviously connected with a real optical lattice.

  1. The authors mis-use the word "ultra-local" in the description of their local critical point. The correlation lengths are several lattice spacings, so they are simply local, not ultra-local .

Report

  1. I must point out again that the Schriefer-Wolff transformation gives additional terms not contained in (2), and despite my remarks in my first report, the authors have not mentioned that they have dropped the co-tunneling terms that hop electrons while flipping the spins on layer two in equation (2). I don’t think the revised paper should be published without a specific acknowledgment, eg “here, we have dropped the three-site terms that arise from properly carrying out the Schrieffer Wolff transformation on the original model” .

  2. The current paper uses J_cs=0.5 - do the authors get the same results if J_cs=0… I suspect not. The authors must be clear about this.

Requested changes

  1. Add acknowlegment to dropped terms in Schrieffer Wolff transformation, eg

“here, we have dropped the three-site terms that arise from properly carrying out the Schrieffer Wolff transformation on the original model” .

  1. The authors use J_cs=0.5. The authors must make it clear whether ultra-local criticality requires a finite J_cs.

  • validity: good
  • significance: good
  • originality: ok
  • clarity: high
  • formatting: reasonable
  • grammar: excellent

Author:  Alexander Nikolaenko  on 2024-06-13  [id 4561]

(in reply to Report 2 on 2024-01-02)

We thank the referee again. We agree with his concerns regarding the three-site hopping terms. To improve the paper, we moved discussion of the optical lattice to the Appendix, and in the main part of the paper, we start directly from the Kondo-Heisenberg model. Furthermore, in the end of Appendix A, we mention that we dropped three-site hopping terms to simplify the analisys.
Regarding his seconds request, we mention in Fig. 7,8,9 that $J_{cs} = 0$, which means that ultra criticality does not require finite $J_{cs}$.

---

## Round 1 · Referee Report · Anonymous (Referee 3) · 2024-3-3

Strengths

1- Solid numerical investigation 2- Timely topic

Weaknesses

Presentation and paper formatting need more work.

Report

The authors investigate a one-dimensional Kondo lattice model using density matrix renormalization group (DMRG) computations and mean-field theory. In a region of the phase diagram they find behavior that they dub "ultra-local criticality" in analogy to phenomena observed in heavy fermions.

The work itself appears to be solid and timely, but presentation needs more work. For example, I was trying to read a printout, but many figures have fonts that were far too small to be readable. Accordingly, I was not able to fully appreciate the work and am afraid that I will have to read again another version where this font-size problem has been fixed.

Nevertheless, I have a few questions concerning content: 1- The authors motivate their work from the heavy-fermion context. However, when they introduce it in Sec. II, they start talking about optical lattices. As it stands, this is a first stumbling stone for the reader. I recommend to either remove the discussion of the optical lattices, at least from the beginning of the paper, or if the optical-lattice context is needed to motivate the model (2) that has actually been investigated, the motivation of the paper has to be rewritten. 2- In the second paragraph of Sec. III A, the authors write "The phase may be labelled as C1S2 or C2S1". If understand correctly and the notation C$n$S$m$ has a well defined meaning, there should be no labeling freedom. 3- In phase II, in particular the red region of Fig. 2, the authors find very puzzling behavior, namely absence of a spin gap, but still a finite (and actually small) spin-spin correlation length $\xi_S$. I believe that there is a possible explanation that the authors have not yet explored, but that they should still discuss: this could simply be a degenerate ground state, maybe due to the open boundary conditions employed in the DMRG simulation. There is a related comment on the finite energy resolution in DMRG in the text below Eq. (F7). Indeed, it is well possible that DMRG is not sufficiently accurate to resolve the small gap in long chains caused by the finite-size splitting of a degenerate ground-state manifold, but that there would be a clear gap if one goes higher up in the spectrum. 4- In the captions of Figs. 4, 7, and 23, the authors refer to a "transfer matrix technique". I believe that this would merit being explained in more detail than just a citation of Ref. [39] in the caption of Fig. 4. 5- I disagree with the statement just before Eq. (3) that "Abrikosov fermion theory" is THE "usual way of dealing with a Kondo lattice model".

Further more specific issues are listed as Requested changes.

Requested changes

1- Make sure that all fonts in all figures are sufficiently large to be readable. As a rule of thumb, no font in a figure should be smaller than the corresponding fonts in the figure caption. 2- Figure captions need to be properly typeset. I suspect that this is a compatibility issue of a package with the style file, but, e.g., Figs. 3, 15, 20, 26, and 27 show that the authors are able to typeset captions properly. 3- Remove the discussion of optical lattices from the definition of the model. 4- On the top left of page 3, there are a few inline equations that are not properly formatted, e.g., "$U>>t$ and $U >> U'$" versus "$U\gg t$ and $U \gg U'$" and "Hubbard U" versus "Hubbard $U$". 5- Top left of page 4: is "expective" really an English word, or do the authors mean "expected"? 6- End of first paragraph of Sec. III A and several instances of the appendices (Fig. 15 and related discussion; appendix F including Fig. 27): there might be confusion between "coherence" and "correlation" lengths. 7- Clarify the phrase "The phase may be labelled as C1S2 or C2S1" in the second paragraph of Sec. III A. Maybe just "The phase could be C1S2 or C2S1"? 8- Explain the "transfer matrix technique" used in Figs. 4, 7, and 23 better than just citing Ref. [39] in the caption of Fig. 4. 9- The letter "$c$" carries two meanings in the present manuscript: central charge $c$ and a velocity $c$. This ambiguity should be removed, probably by changing the letter used for the velocity at the beginning of page 7 and at the top of page 14. 10- The caption of Fig. 7 is disproportionally long. It would probably be better to move some of the discussion / explanation into the main text. 11- At the beginning of page 11, the authors write "In the appendix, ...". There are several appendices; hence they should specify which one they are referring to. 12- Revisit the statement "The usual way of dealing with a Kondo lattice model is through Abrikosov fermion theory" just before Eq. (3). 13- Why do Refs. [25,27,39] have an additional arXiv preprint number? In Ref. [28] it's just a duplicate arXiv preprint number. 14- Eqs. (C4)-(C6) and (C8)-(C10): Is the subscript $_i$ and the factor $i$ in front of the integral the same "$i$"? 15- The color contrast of some labels in Fig. 21 is bad: in particular a red "$J_K = \ldots$" on a dark blue background is very difficult to read. 16- Last line of appendix E: the closing "$\vert$" of the absolute value is missing. 17- Second line of caption of Fig. 27: What is the "$\xi^-1$"?

  • validity: high
  • significance: high
  • originality: high
  • clarity: good
  • formatting: below threshold
  • grammar: reasonable

Author:  Alexander Nikolaenko  on 2024-05-21  [id 4504]

(in reply to Report 3 on 2024-03-03)
Category:
objection
reply to objection

"The authors investigate a one-dimensional Kondo lattice model using density matrix renormalization group (DMRG) computations and mean-field theory. In a region of the phase diagram they find behavior that they dub "ultra-local criticality" in analogy to phenomena observed in heavy fermions.

The work itself appears to be solid and timely, but presentation needs more work. For example, I was trying to read a printout, but many figures have fonts that were far too small to be readable. Accordingly, I was not able to fully appreciate the work and am afraid that I will have to read again another version where this font-size problem has been fixed."

Reply: We thank the referee for accurately reading and summarizing the paper and generally positive feedback.

1) " The authors motivate their work from the heavy-fermion context. However, when they introduce it in Sec. II, they start talking about optical lattices. As it stands, this is a first stumbling stone for the reader. I recommend to either remove the discussion of the optical lattices, at least from the beginning of the paper, or if the optical-lattice context is needed to motivate the model (2) that has actually been investigated, the motivation of the paper has to be rewritten."

Reply: Our primary motivation for studying the model was to understand the transition between the Luther-Emery liquid and Luttinger liquid: in particular the critical regime. However, we also wanted to draw a connection between the theory and experiment. That is why we proposed to simulate our model using recent advances in optical lattices in Section II.

2) " In the second paragraph of Sec. III A, the authors write "The phase may be labelled as C1S2 or C2S1". If understand correctly and the notation CnSm has a well defined meaning, there should be no labeling freedom."

Reply: We have not been able to resolve fully whether this phase is C1S2 or C2S1. However, we made a conjecture in the next sentence: "We conjecture that it is C2S1 and there is only one spin mode, given that the peak of the spin-spin correlation function seems to be pinned at 2kF ". But we agree with the referee that the wording was poor so we change the sentence to "Based on the value of the central charge, the phase could be either C1S2 or C2S1".

3) " In phase II, in particular the red region of Fig. 2, the authors find very puzzling behavior, namely absence of a spin gap, but still a finite (and actually small) spin-spin correlation length $\xi_S$. I believe that there is a possible explanation that the authors have not yet explored, but that they should still discuss: this could simply be a degenerate ground state, maybe due to the open boundary conditions employed in the DMRG simulation. There is a related comment on the finite energy resolution in DMRG in the text below Eq. (F7). Indeed, it is well possible that DMRG is not sufficiently accurate to resolve the small gap in long chains caused by the finite-size splitting of a degenerate ground-state manifold, but that there would be a clear gap if one goes higher up in the spectrum."

Reply: Indeed, highly degenerate ground state, or flat band, could explain many of the observed quantities: in particular vanishing spin gap with the finite coherence length. We explored this scenario both numerically and theoretically. In Fig. 9c) we computed the dynamical spin structure factor and showed that there are many gapless spin fluctuations in a range of momentum. In Appendix F, we demonstrated how a nearly flat band can lead to a finite coherence length above some finite temperature. However, the true origin of this ultra-critical region remains unknown to us.

4) " In the captions of Figs. 4, 7, and 23, the authors refer to a "transfer matrix technique". I believe that this would merit being explained in more detail than just a citation of Ref. [39] in the caption of Fig. 4."

Reply: Transfer matrix technique is the standard technique in dealing with infinite systems and we decided not to over-complicate the paper by describing how both algorithms(DMRG and iDMRG) work. Besides, the extraction of the entanglement entropy is well documented in the original TenPy package.

5) " I disagree with the statement just before Eq. (3) that "Abrikosov fermion theory" is THE "usual way of dealing with a Kondo lattice model".

Reply: We agree with the referee and thus changed the sentence: " One of the ways in dealing with a Kondo lattice model is through the Abrikosov fermion theory".

List of changes: We accounted for most of the requested changes by the referee in the resubmitted version of the paper.

---

## Round 1 · Author Response

Warnings issued while processing user-supplied markup:

  • Inconsistency: plain/Markdown and reStructuredText syntaxes are mixed. Markdown will be used.
    Add "#coerce:reST" or "#coerce:plain" as the first line of your text to force reStructuredText or no markup.
    You may also contact the helpdesk if the formatting is incorrect and you are unable to edit your text.

" In this paper, the authors carry out an extensive DMRG study of a one dimensional Kondo lattice model derived from a two-chain Hubbard model. The authors have discovered two novel phases that appear to develop between the Luther-Emery insulator and the large Fermi surface Luttinger Liquid; moreover, the associated quantum phase transition displays the characteristic of a local quantum phase transition. This is a fascinating result that deserves publication in scipost. I do however have a number of questions for the authors that require addressing before the paper is pubished. Question (1) below is crucial. "

Reply: We thank the referee for carefully reading our paper, for the concise summary of our work and generally positive feedback.

" (1) The derived Kondo lattice model in equation (2) is missing the cotunneling terms that describe the simultaneous "hop and spin-flip" processes that develop when an electron on layer 1 undergoes a virtual transition into layer 2 and back again. In the authors work, they have assumed that when this process occurs, the electron always returns to the same site on chain 1. However, it can return to its neighboring, or next neighboring site. The authors need to carefully repeat their Schrieffer Wolff transformation to include this effect. These terms should be included in (2). (See eg Phys. Rev. B 57, 12757 (1998)). "

Reply: We thank the referee for raising this issue, which is certainly a valid point. Indeed, $t_{12}$ term also produces three site terms which were omitted in the following paper. They are given, for example, in the previous paper of one of the authors, see Phys. Rev. B 106, 045103 (2022), Eq. A3. These terms are quite cumbersome to account and even without them, the model given by Eq. 2 appeared to be quite interesting. Furthermore, it was demonstrated that the terms of a similar nature with $J_{cs} $ coupling did not change the physics much, see the discussion below.

" The model Eq (2) is interesting in its own right, but its motivation and potential link with an optical lattice is slightly blurred without the link with the two-chain model of equation (1). Can we be sure that the misssing cotunneling terms in equation (2) are restored, the phase diagram is unchanged? This is a crucial question for the paper. I recognize that this would require more computation, but without it, the results of the paper are only valid for the two chain model with $t_{12}=0.$ "

Reply: It was rightfully noted by the referee that the model in Eq. 2 is interesting in its own right. In fact, the main goal of the work was to study the phase transition between Luther-Emery liquid and Luttinger liquid with large Fermi surface, in particular the critical regime with ultra-quantum criticality. Most probably, missing terms change the boundaries of the phases, but will not change the phases themselves. In the real optical lattice experiment, it is actually hard to generate a finite $t_{12}$, so our results at $t_{12}=0$ is directly relevant to the experiment.

" (3) If the authors do not wish to do the extra runs, then can we at least be sure that the physics is the same for $J_{cs} = 0$, i.e $t_{12}=0$? I note that region II was studied at $J_{cs} =0$, so this may be fine, but region II was studied at finite $J_{cs}$, so it is not clear whether these results will hold at $J_{cs} =0$, or with the extra co-tunneling. "

Reply: We studied the model for both $J_{cs}=0,0.5$ and discovered that all interesting physics remains the same. The LL,LE, phase 1 and phase 2 are still present, and turning on $J_{cs}$ only slightly($\delta J_K\approx 0.05$) shifts the boundaries of the phases on the phase diagram. In fact, even for Kondo-Heisenberg Hamiltonian, see Eq. E1, the Ultra-critical regime was present. The phase 1 for $x=0.25$ just shifts from $J_K=(2.1-2.5)$, see Fig. 2, to $J_K=(2.5-2.8)$, see Appendix E discussion. Therefore, we think that other missing terms coming from $t_{12}$ interactions would not make a big difference. However, if one needs to locate the precise positions of the boundaries, these terms must be taken into account. This could be the part of the future work, to make it closer to the experimental realizations. Our inclusion of a finite $J_{cs}$ is not essential.

" (4) In region I, how do you rule out the possibility that the spin gap is just too small to be resolved - i.e can you rule out the possibility that the c=3 region is just a cross-over that preceeds the transition to the LE liquid? "

Reply: For region 1, we computed the spin gap for different length sizes up to $L=155$ and showed that it scales as $1/L$ to zero. If there was a spin gap, the quantity $\Delta_s L$ would increase as we increase $L$, see Fig.4a at $J_K<2$. Moreover, the central charge is increased to 3, meaning that there are additional gapless modes.

" (5) In region II, I was not clear what the authors meant by a ferromagnetic moment. Normally we reserve this description for a system with a finite moment per unit length, yet the authors claim on p9, column 1, that the total moment $S_z$ is 1\% of (1/L). I suspect this is a typo - do the authors mean M = (Sz/L) = 1\% (1/L), i.e $S_z ~ 1\%$. On page page 10, column 1 says M ~ 1\%. This is confusing! Which is the right claim? Is there a bulk finite M, or is M~ 1\% (1/L)? "

Reply: In the paper, we meant $M=S_z^{tot}/L=0.01$. We changed the corresponding sentences.

" (6) What do the authors mean by "ultra-local"? The correlation lengths appear to be of order 1-5. Surely, "local" is sufficient? "

Reply: We called that regime ultra-local, to distinguish it from some previous literatures which claim local criticality for a finite $z$. The name 'ultra-local' criticality is first mentioned by the paper of Else and Senthil, see PRL 127, 086601 (2021).

---

## Round 1 · List of Changes

Deleted (1/L) in the sentence "However, here we find that the FM moment is very small and only at the order of 1$\%$ (1/L)."

---

## Round 2 · Referee Report · Anonymous (Referee 3) · 2024-6-9

Report

With their revised version, the authors have made some improvements, but I was surprised how many important points have not been addressed. For example, I tried to read a revised version, but still had the same font-size issues as in the previous round. Thus, I am afraid that we have not really advanced much since the previous round of review.

In addition, I stumbled across some further minor issues when re-reading the manuscript. See "Requested changes" for further details.

Requested changes

1- The font-size issue with the figures still does need to be addressed. For example, the insets of Fig. 5 look like rescaled full-size figures. However, when rescaling figures, one needs to increase the font size and line widths accordingly. I am afraid that such issues still concern a big fraction of the figures. Maybe some of these would become obsolete during production, but typesetting the manuscript with the SciPost Physics template (see https://scipost.org/SciPostPhys/authoring#manuprep) would help to check. 2- Red text on a dark blue background is a bad idea (see, e.g., Fig. 14), in particular if in addition fonts are tiny. 3- The motivation for this work is still from heavy fermions, but when the authors introduce the model, they change their mind and motivate it from optical lattices. It is fine to appeal to additional motivation, but the transition needs to be made smoother. 4- I believe that the Requested changes in Anonymous Report 2 on 2024-1-2 have been simply ignored. 5- "Transfer matrix technique": maybe it is standard and the method can just be referenced from the literature. However, I do not think that one should introduce new notations in figure captions (specifically here the caption of Fig. 4; note that the term appears again at the bottom left of page 7 and in the caption of Fig. 23). 6- I believe that the label of the horizontal axes of Figs. 4(c,d) should read '$q$' rather that '$x$'. 7- Ref. [11] appears to have been published in Science 377, 1-10 (2022). 8- Ref. [28] was in the meantime published in Phys. Rev. Research 6, 023227 (2024). 9- There are a number of minor typographical issues: a) Systematic absence of spaces before the \cite command. b) Missing '$\rangle$' in the definition of $V$ on the third line of the caption of Fig. 3. c) Strange typesetting of '$J_K$' seven lines below Fig. 3. d) Missing full stop in '... compressibility $\kappa^{−1}$. In the insulating ...' at the beginning of section III C. e) Spurious 'to' in 'with to two spin-gapped phases' at the beginning of section IV A. f) Left column of page 12: Do the authors really mean 'optical' or rather 'optimal' 'doping'? There are a few further minor typographic and grammatical errors (such as missing articles) that I refrain from listing. However, I believe that the manuscript would benefit from careful proofreading, ideally from a native English speaker.

Recommendation

Ask for major revision

  • validity: high
  • significance: high
  • originality: high
  • clarity: good
  • formatting: below threshold
  • grammar: reasonable

Author:  Alexander Nikolaenko  on 2024-06-13  [id 4560]

(in reply to Report 1 on 2024-06-09)

We thank the referee again, most of their objections are valid and below we present a list of changes we made as requested by the referee.

1) We increased fonts of the insets of Fig. 5.

2) We changed the red color of the Fig. 14 and Fig. 21 to the light green, which is easier to read on blue and increased fonts.

3) We moved the discussion of optical lattices to the Appendix A, now we start directly from the Kondo-lattice model and discuss the obtained phase diagram.

4) We apologize for ignoring the Anonymous Report 2 on 2024-1-2 which we missed. We answered it now. We added the sentence saying that we dropped three-site terms in the end of the appendix. We also stress that ultra-criticality does not require a finite $J_{cs}$ as can be seen in Fig.7,8,9 for example.

5) We moved the discussion of how we obtain the central charge from the captions of Figures 4, 7, 23 to the main text.

6) We have changed labels for Figure 4c),d).

7,8) We updated these references.

9) We corrected these minor typographical mistakes suggested by referee.

---

## Editorial Decision

published